# Transcriptional and genomic parallels between the monoxenous parasite Herpetomonas muscarum and Leishmania

**Megan A. Sloan**[1], **Karen Brooks**[2], **Thomas D. Otto**[2¤], **Mandy J. Sanders**[2], **James A. Cotton**[2]*, **Petros Ligoxygakis**[1]*

**1** Department of Biochemistry, University of Oxford, Oxford, United Kingdom, **2** The Wellcome Sanger Institute, Wellcome Genome Campus, Hixton, Cambridgeshire, United Kingdom

¤ Current address: Centre of Immunobiology, Institute of Infection, Immunity and Inflammation, College of Medical, Veterinary and Life Sciences, University of Glasgow, Glasgow, United Kingdom
* jc17@sanger.ac.uk (JAC); petros.ligoxygakis@bioch.ox.ac.uk (PL)

**Data Availability Statement:** All sequencing data are available at the European Nucleotide Archive (ENA) under accession number ERP008869.

## Abstract

Trypanosomatid parasites are causative agents of important human and animal diseases such as sleeping sickness and leishmaniasis. Most trypanosomatids are transmitted to their mammalian hosts by insects, often belonging to Diptera (or true flies). These are called dixenous trypanosomatids since they infect two different hosts, in contrast to those that infect just insects (monoxenous). However, it is still unclear whether dixenous and monoxenous trypanosomatids interact similarly with their insect host, as fly-monoxenous trypanosomatid interaction systems are rarely reported and under-studied–despite being common in nature. Here we present the genome of monoxenous trypanosomatid *Herpetomonas muscarum* and discuss its transcriptome during *in vitro* culture and during infection of its natural insect host *Drosophila melanogaster*. The *H. muscarum* genome is broadly syntenic with that of human parasite *Leishmania major*. We also found strong similarities between the *H. muscarum* transcriptome during fruit fly infection, and those of *Leishmania* during sand fly infections. Overall this suggests *Drosophila-Herpetomonas* is a suitable model for less accessible insect-trypanosomatid host-parasite systems such as sand fly-*Leishmania*.

## Author summary

Trypanosomes and *Leishmania* are parasites that cause serious Neglected Tropical Diseases (NTDs) in the world's poorest people. Both of these are dixenous trypanosomatids, transmitted to humans and other mammals by biting flies. They are called dixenous as they can establish infections in two different types of hosts– insect vectors and mammals. In contrast, monoxenous trypanosomatids usually only infect insects. Despite establishment in the insect's midgut being key to transmission of NTDs, events during early establishment inside the insect are still unclear in both dixenous and monoxenous parasites. Here, we study the interaction between a model insect–the fruit fly *Drosophila melanogaster*–and its natural monoxenous trypanosomatid parasite *Herpetomonas muscarum*. We show that both the genome of this parasite, and gene regulation at early stages of infection

**Funding:** KB, TDO, MJS and JAC were supported by Wellcome via their core support for the Wellcome Sanger Institute (WSI) through grant 206194. Work in Oxford was supported by a Consolidator grant from the European Research Council (310912 Droso-Parasite, to PL), project grant BB/K003569 from the Biological and Biotechnological Sciences Research Council (to PL) and a Wellcome Trust doctoral scholarship (to MAS). The funders had no role in study design, data collection and analysis, decision to publish, or preparation of the manuscript.

**Competing interests:** The authors have declared that no competing interests exist.

have strong parallels with *Leishmania*. This work has begun to identify evolutionarily conserved aspects of the process by which trypanosomatids establish in insects, thus potentially highlighting key checkpoints necessary for transmission of dixenous parasites. In turn, this might inform new strategies to control trypanosomatid NTDs.

## Introduction

The family Trypanosomatidae belong to the order Kinetoplastida, a group characterized by the presence a mitochondrial organelle rich in DNA (kDNA) called the kinetoplast. This family includes parasitic flagellates that undergo cyclical development in both vertebrate and invertebrate hosts (and are therefore dixenous). These parasites are best known as agents of important diseases in humans, domestic animals and plants. However, several genera of this order such as *Crithidia*, *Herpetomonas*, *Blastocrithia* and *Leptomonas* are restricted to a single host (monoxenous), usually an insect from the orders Diptera, Hemiptera or Siphonaptera [1]. Although such monoxenous or "lower" trypanosomatids seem to have their lifecycle essentially confined to insect hosts [2], they have also been reported in plants [3] and immunocompromised humans [1].

There is an increasing interest in monoxenous trypanosomatids as a model for understanding the evolution and ecology of trypanosomatids [4], as well as how they may modify their insect host [4]. It is now clear that monoxenous trypanosomatids are ubiquitous parasites of a wide range of insect groups and have numerous effects on the physiology of the insect host. These effects include alterations in fertility and reproduction, modified food intake, delayed development and reduction in lifespan [5]. In projections of total animal biodiversity, insects represent more than 60% of all animals [6]. Therefore, knowledge of insect physiology and what can influence it, is essential for maintaining a species-rich environment especially when longitudinal population data show a sharp decline in flying insect biomass [7]. In this context, studies of trypanosomatid-insect interactions will provide vital insights into the ecology of crucial insect species (e.g. pollinators).

To this end, a number of monoxenous trypanosomatid genomes and transcriptomes are being investigated [8,9]; including bee parasites from the genus, *Lotmaria passim* (the honey bee parasite) and *Leptomonas pyrrhocoris* a globally disseminated parasite isolated from fire bugs [10,11]. These studies, and earlier work on the molecular biology of trypanosomatids, have revealed that monoxenous parasites share many distinctive genome features with their better-studied dixenous relatives [12].

The genomic DNA is arranged into 'polycistronic' (multi-gene) transcriptional units of functionally unrelated genes, the majority of which lack introns. Given this gene arrangement, the cells do not control an individual gene's expression by varying its transcription level, instead expression is controlled by RNA-binding proteins [13] and other post-transcriptional processes such as RNA editing [14]. RNA editing processes include trans-splicing where 39 nucleotides, called a splice leader sequence, are added to the 5' end of mRNAs [15]. The splice leaders (also called mini exons) are encoded in tandem repeats in a different genomic locus to the gene.

Trypanosomatid kDNA is arranged in interlocking 'maxi-circles' [16–18]. The kDNA maxicircle is homologous to mitochondrial genomes in other systems but the sequence encoding many of typical mitochondrial proteins is scrambled, relying on post-transcriptional mRNA editing to reconstitute the correct coding sequence [19]. The kinetoplast also contains

thousands of associated 'mini-circles' which encode guide RNAs involved in this editing process [17].

In addition to ecological insights, studies of monoxenous trypanosomatids may help us gain new perspective on interactions of more medically important parasites and their insect vectors, which mediate neglected tropical diseases such as Leishmaniasis (vectored by phlebotomine sand flies) and sleeping sickness (tsetse flies). To inform, and accelerate, research in these experimentally challenging dipteran-parasite relationships, we have developed the study of the model dipteran *Drosophila melanogaster* and its natural trypanosomatid *Herpetomonas muscarum* [20]. We have established that a network of signalling in the intestine of the host was important for clearance as well as for maintaining fecundity. This network involved NF-κB and STAT-mediated transcription, which regulate intestinal stem cell proliferation that the parasite attempts to suppress. Here, we turn our attention to the parasite. We report the genome of *H. muscarum* isolated from a wild population of *Drosophila melanogaster* in Oxfordshire, UK. We also report the transcriptomes of this *H. muscarum* isolate from *in vitro* culture and during the course of infection in *D. melanogaster*. The similarities with *Leishmania major* both at the genome level as well as transcriptome regulation were striking. This was especially the case in the early phases of host infection when the parasite needs to overcome the barrier of the insect midgut and establish infection. Given the resistance mechanisms to parasite establishment (and therefore onward transmission) reside in the dipteran midgut [21], the *Drosophila-Herpetomonas* model may allow researchers to take advantage of the extensive toolkit of genetic approaches available for *Drosophila* to uncover mechanistic details of evolutionary conserved aspects of the relationship between trypanosomatids and dipteran vectors, where the tool-box for functional studies is not yet fully developed.

## Results/Discussion

### The *Herpetomonas muscarum* genome

**Assembly.**  PacBio and Illumina sequence reads were generated from an axenic culture of *H. muscarum* promastigotes as described in Materials and Methods. The reads were assembled into a genome of 41.7 Mbp in 264 scaffolds with the largest 1,793,442 bp in length (N50 = 707,495 bp). We observed a median read coverage of 114x with populations of scaffolds coverage at approximately 50x and 160x which may represent monosomic and trisomic scaffolds (Fig 1, predicting 37–39 chromosomes). Kmer analysis of the sequencing reads estimated the haploid genome length to be approximately 35.2 Mbp with a read error rate of less than 1% (S1 Fig, Vurture *et al.*, 2017). While the GenomeScope [22] model does not fit the aneuploid nature of trypanosomatid genomes (see below), we believe this suggests our assembly is approximately the correct size.

**Annotation.**  Gene model annotation was generated with Companion [23] using evidence from RNA-seq data (described below) and the proteomes of *L. major*, *L. braziliensis* and *T. brucei* as described in Materials and Methods. The final *H. muscarum* v1 annotation contains 12,687 genes, of which 12,162 are inferred to be protein-coding (Table 1).

All unique open reading frames produced by the gene models were kept, even in cases where the gene prediction was not strongly supported by RNA-sequencing evidence, in an attempt to not 'miss' genes. It is therefore likely that this annotation contains a higher number of genes than the 'true annotation'. However, the number of reported genes is close to that reported for other trypanosomatid species e.g. *T. brucei* TREU927 strain contains 11,567 genes [24]. We also note that the few T. brucei genes reported to contain intronic sequences, e.g. poly(A)-polymerase (Tb927.3.3160) and the mini-exon gene (see below), also appear to contain intronic regions in *H. muscarum*.

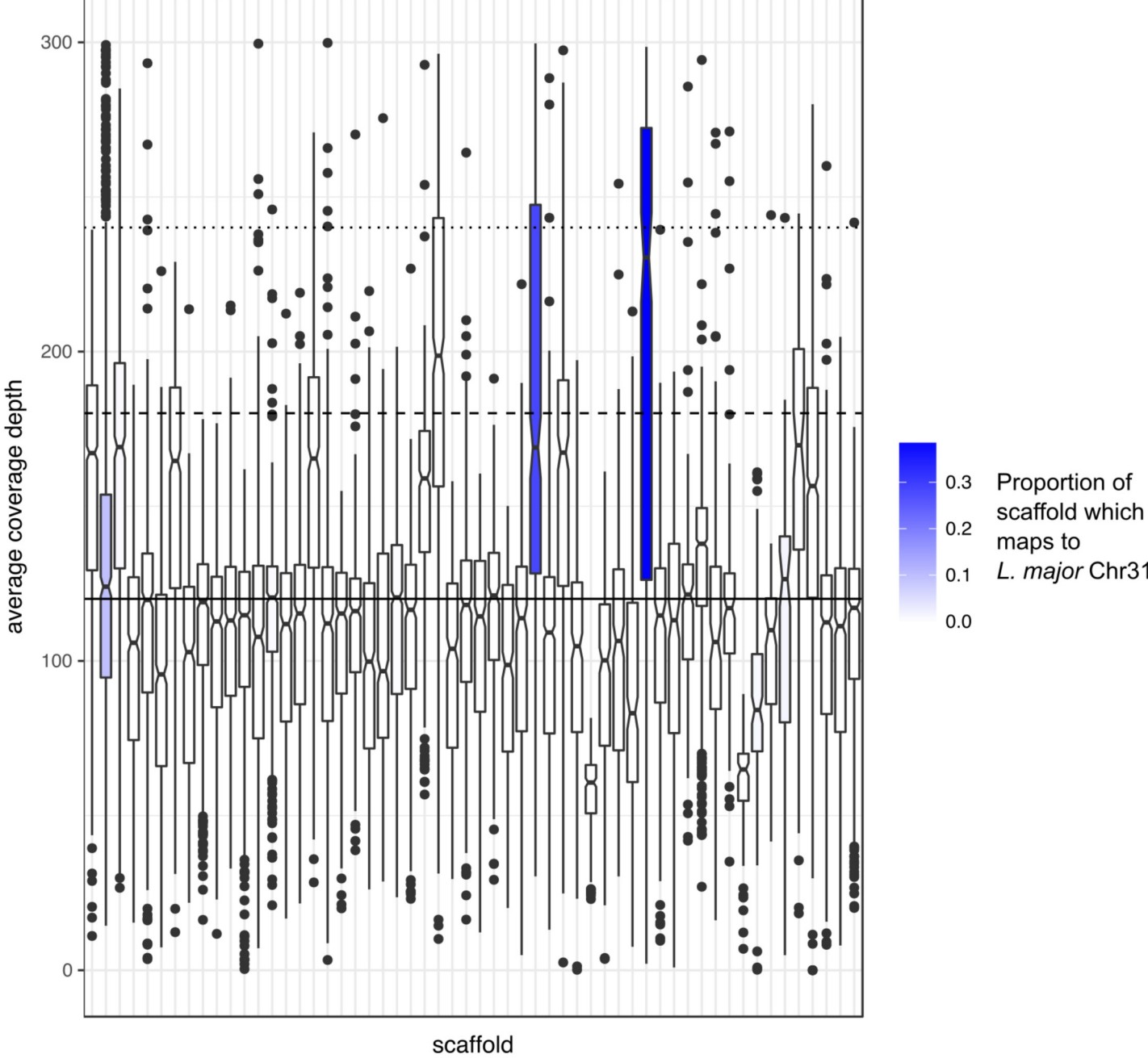

**Fig 1. Average coverage depth of *H. muscarum* scaffolds > 100 kb.** The solid line shows the global median read coverage. The dashed line shows 1.5x and the dotted line shows 2x the global median read coverage respectively. In blue are scaffolds which were mapped, by PROmer (Kurtz *et al.*, 2004), to the *L. major* chromosome 31—sequences of 300bp which map with > 70% identity. The shade of blue represents the proportion of the scaffold which was mapped.

### Conserved features of trypanosomatid genomes

**Genome structure and large scale synteny.** As seen in other trypanosomatid genomes, open reading frames were found on both strands on many scaffolds. Genes are (mostly) arranged in large groups of genes present on the same strand and in the same direction, which is indicative of the polycistronic transcripts typical in trypanosomatid genomes. The regions between polycistrons, commonly referred to as strand switch regions (SSRs), are thought to

**Table 1.** *Herpetomonas muscarum* genome annotation summary.

| Feature | *H. muscarum* v1.0 |
|---|---|
| Genes | 12687 |
| mRNAs | 12162 |
| CDSs | 12175 |
| Polypeptides | 12934 |
| Pseudogenes | 772 |
| rRNAs | 168 |
| snRNAs | 3 |
| snoRNAs | 181 |
| tRNAs | 173 |

contain the transcriptional start sites for transcription of each group of genes. We used the SSRs to define and estimate the number of polycistrons. Here we defined SSRs to begin and end at genes where the downstream open reading frame is on the opposing strand of the same scaffold. This highlighted 386 genes from 112 different scaffolds. These putative strand switches were manually inspected and could be grouped into different three situations. There were 128 *bona fide* strand switches which were either divergent (72 cases) or convergent (56 cases) (S1 Table). There were 166 cases where a single gene (or small group of < 5 genes) had become inverted within a polycistron. Small genes (< 350bp) encoding hypothetical proteins and tRNAs were commonly found in these cases, though other larger genes were also found in these groups e.g. HMUS00935500.1 an putative trans-sialidase. Finally, there were 92 cases where a strand switch does occur, but the precise locus was unclear. These cases tended to be at where a single gene at the end of a scaffold was on the opposing strand to all other genes on the scaffold–as such it was unclear if this represented a *bona fide* strand switch or a single gene inversion. Overall, this indicated there are at least 128 polycistrons in the *H. muscarum* genome, though this is likely to be an underestimate given the ambiguity of some strand switch regions. Comparisons with other trypanosomatids genomes also suggest this figure is an underestimate, e.g. *L. major* is predicted to have 184 polycistrons [25] and *T. brucei* is predicted to have 150 [26], both of which have smaller genomes and fewer predicted chromosomes than *H. muscarum*.

Despite diverging before the existence of mammals [27], trypanosomatids show high gene order conservation across the genome. As expected, the *H. muscarum* scaffold showed synteny with other trypanosomatid genomes (Fig 2A–2E). *Herpetomonas* was most highly syntenic with *L. major* despite being considered phylogenetically closer to *Phytomonas* and *Leptomonas*. To quantify this, we took non-overlapping windows of adjacent *H. muscarum* genes with single copy orthologs in three comparator genomes: *L. major*, *T. brucei* and *Leptomonas seymouri*. For each window size, we count for how many windows have all orthologs on the same scaffold in the comparator (syntenic windows), and for how many of those all the genes are in the same relative order as their *H. muscarum* orthologs (colinear windows). Almost 96% of 3-gene windows of single-copy orthologs between *H. muscarum* and *L. major* (1845/1926) are syntenic, and 53% of these are colinear (985/1845). This conserved genome structure is shared, to a slightly lesser extent across the trypanosomatids (91.7% or 1386/1511 syntenic with *T. brucei brucei*, 55% or 766/1386 colinear, 80.9% or 1643/2030 syntenic with *L. seymouri*, 46% or 761/1643 colinear). This relationship holds across window sizes (Fig 2F). The values for synteny with *Leptomonas seymouri* are likely to be biased downwards by the fragmentary assembly available for that species, and this analysis does not capture rearrangements, expansions or contractions of multi-gene families, for which one-to-one orthology is unlikely to be clear.

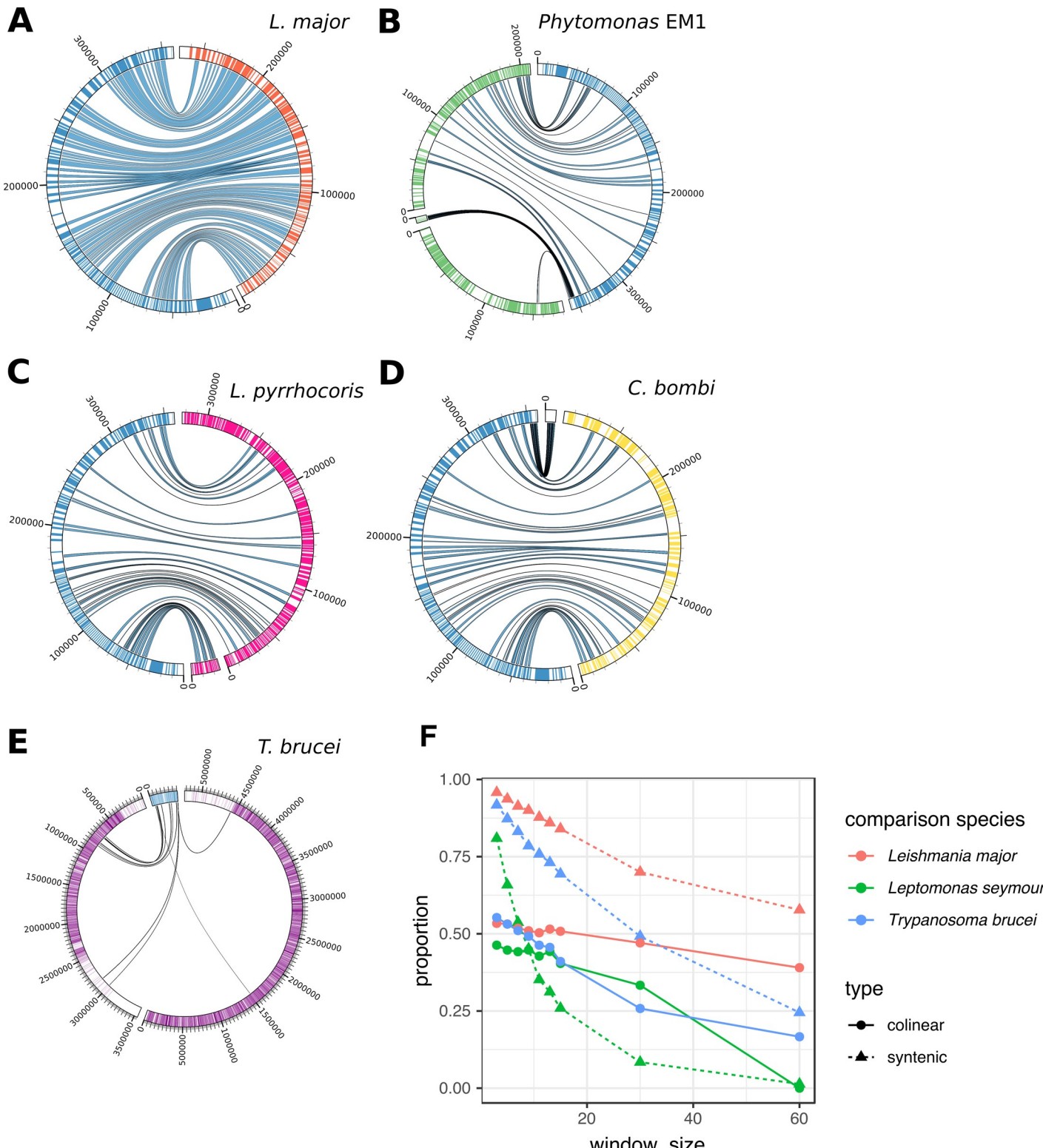

**Fig 2. Synteny and colinearity between *H. muscarum* and other trypanosomatids.** As an example, this plot shows co-linearity between *H. muscarum* genes (genes highlighted in blue) on scaffold 40 and: **A.** *L. major* chromosome 1 (genes highlighted in red). **B.** *Phytomonas EM1* scaffolds HF955082, HF955140 and HF955140 (genes highlighted in green) **C.** *Leptomonas pyrrhocoris* scaffolds LpyrH10_33 and LpyrH10_41 (genes highlighted in pink) **D.** *Crithidia bombi* scaffolds (genes highlighted in yellow) OESO01000125 and OESO01000148. **E.** *Trypanosoma brucei* chromosomes 9 and 11 (genes highlighted in purple). Scaffold/Chromosome labels show length in bp. This data was produced using Promer alignments (Delcher *et al.*, 2002). Ribbons between scaffolds show windows of >100 amino acid (translated)

align with at least 50% identity. This data was visualised using Circos (Krzywinski et al. 2009). To quantify these relationships, we investigated all windows of consecutive genes with single-copy orthology in *H. muscarum* in comparison to *L. major*, *T. brucei brucei* and *Leptomonas seymouri*. **F.** Shows the proportions of these windows for which all genes occurred on a single scaffold in the comparison genome (syntenic windows), and the proportion of those for which all gene occurred in the same order as in *H. muscarum* (colinear windows) for a range of window sizes from 3 to 60 genes. Numbers of windows included in the comparisons varies from 1926 windows of 3 single-copy orthologs with *L. major* to 49 windows of 60 adjacent genes with single-copy orthologs in *T. brucei*. Note that synteny values are also affected by the degree of continuity of the comparison species genome for *Leptomonas seymouri*.

**Splice leader sequence.** In trypanosomatids, each mRNA is capped, via trans-splicing [reviewed in 15], with a conserved 39bp sequence called the splice leader (SL). The SL is encoded by the mini-exon genes which are found throughout the genome in tandem arrays. Each mini-exon has two components; the highly conserved 39bp sequence trans-spliced on to mRNAs (the exon) and a less well conserved intronic sequence. Between each mini-exon gene there is a variable spacer region which is not transcribed. To find the splice leader sequence for our *H. muscarum* isolate, we searched for the conserved 39bp SL sequence from *Phytomonas serpens* (L42381.1) in the *H. muscarum* scaffolds. This gave 259 hits over 24 scaffolds, which we used to identify 19 clusters of mini-exon gene repeats (over 15 scaffolds) containing 3–43 copies of the mini exon gene (see S2 Table). The first 111bp of the gene are common to all copies of the mini-exon gene and contain a 40bp splice leader sequence and what we predict to be the intron.

The splice leader sequence (1-40bp) and the putative intronic region (41-111bp) were then aligned with mini-exon sequences of several other trypanosomatids in the Leishmaniinae clade—including 9 other *Herpetomonas* isolated from heteropterans in the neotropics [28]. Whilst the splice leader sequence is well-conserved across the clade (Table 2), we observe variability in the A/T-rich region between bases 11-19bp which appears genus specific, with the exception of the *Herpetomonas* sequences. *H. rotimani* and *H. nabiculae* have identical sequence across the 11-19bp region. However, the *H. muscarum* and *H. nabiculae* differ from each other, and the other *Herpetomonas* sequences over this variable region. Additionally, compared to other trypanosomatids, the *Herpetomonas* sequences have an 'additional' adenosine between bases 10 and 11. The intronic region from *H. muscarum* shows high similarity to that of previously reported *Herpetomonas* sequences. The first 15bp of the intronic sequence appear to be conserved in other species from the Leishmaniiae clade, however the sequence becomes more variable thereafter in both in terms of base content and length.

**Tubulin loci.** The architecture of the tubulin arrays has been described in a number of trypanosomatids [29], with two mutually exclusive formats being defined–monotypic and alternating. Monotypic tubulin arrays consist of either alpha-tubulin or beta-tubulin. Alternating arrays contain both alpha-tubulin and beta-tubulin genes which alternate along the array. The *H. muscarum* orthologues of *Trypanosoma brucei* alpha and beta tubulin genes were found using Orthofinder and used to locate the tubulin arrays.

We identified three genomic loci containing *H. muscarum* tubulin genes (Fig 3). Two of these loci consist of beta-alpha alternating arrays and the third locus consists of four copies of a beta tubulin genes. The alternating beta-alpha arrays are consistent with previous findings (reported as *Herpetomonas megaseliae*) [29] and suggested that, like *T. brucei*, *H. muscarum* genome has the alternating tubulin array configuration. However, the presence of a monotypic beta tubulin array in addition to the alternating arrays contrasts the established model in which each species has either alternating or monotypic arrays, but not both.

The genes surrounding the monotypic beta tubulin locus shared some synteny with regions of chromosome 4 of *T. brucei* and chromosome 8 of *L. major* (gene numbers Tb927.5.970 – Tb927.927.5.3090 and Lmj.08.1090-Lmj.08.11140). Interestingly this region of *L. major* chromosome 8 is one of two singleton beta-tubulin loci in the species. As such, the tubulin

**Table 2. Alignment of highly conserved splice leader sequences (bases 1–40 of mini-exon gene) of *H. muscarum* and other species from the Leishmaniiae clade.** The variable AT-rich region (positions 11–19, bold) is shown by genus. *Herpetomonas sp.* appear to have an additional A or T residue, dependant on species at position 11.

| Species | Accession # | Splice leader sequence (bases 1–40) |
|---|---|---|
| *Herpetomonas muscarum* | | AACTAACGCT**AAAAATTGTT**ACAGTTTCTGTACATTATTG |
| *Herpetomonas muscarum* | EU095982.1*, EU095980.1*, EU095979.1*, EU095983.1, EU095984.1, EU095981.1* | AACTAACGCT**AAAAATTGTT**ACAGTTTCTGTACTATATTG |
| *Herpetomonas sp. TCC263* | EU095976.1 | AACTAAAGCA**TTATATAGAT**ACAGTTTCTGTACTATATTG |
| *Herpetomonas sp. TCC263* | EU095977.1 | AACTAAAGCA**TTATATAGAT**ACAGTTTCTGTACTATATTG |
| *Herpetomonas roitmani* | EU095978.1 | AACTAAAGCA**TTATATAGAT**ACAGTTTCTGTACTTTATTG |
| *Herpetomonas nabiculae* | KF054153.1 | AACTAACGCTA**T-TATTGTT**ACAGTTTCTGTACTTTATTG |
| *Phytomonas EM1* | X87138.1 | AACTAACGCT-**ATTCTAGAT**ACAGTTTCTGTACTTTATTG |
| *Phytomonas serpens* | L42381.1, L42378.1, L42377.1, L42382.1, L42376.1 | AACTAACGCT-**ATTCTAGAT**ACAGTTTCTGTACTTTATTG |
| *Phytomonas sp. Mar8* | AF250993.1 | AACTAACGCT-**ATTCTAGAT**ACAGTTTCTGTACTTTATTG |
| *Phytomonas sp. Alp1* | AF250967.1 | AACTAACGCT-**ATTCTAGAT**ACAGTTTCTGTACTTTATTG |
| *Leishmania braziliensis* | MG010484.1 | AACTAACGCT-**ATATAAGTA**TCAGTTTCTGTACTTTATTG |
| *Leishmania tarentolae* | AY100201.1 | AACTAACGCT-**ATATAAGTA**TCAGTTTCTGTACTTTATTG |
| *Leishmania hoogstraali* | AY100197.1, AY100200.1 | AACTAACGCT-**ATATAAGTA**TCAGTTTCTGTACTTTATTG |
| *Leishmania gymnodactyli* | AY100195.1, AY100196.1 | AACTAACGCT-**ATATAAGTA**TCAGTTTCTGTACTTTATTG |
| *Leishmania adleri* | AY100199.1, AY100194.1 | AACTAACGCT-**ATATAAGTA**TCAGTTTCTGTACTTTATTG |
| *Leishmania major* | XR_002460055.1 | AACTAACGCT-**ATATAAGTA**TCAGTTTCTGTACTTTATTG |
| *Leishmania mexicana* | Agami and Shapira 1992 | AACTAACGCT-**ATATAAGTA**TCAGTTTCTGTACTTTATTG |
| *Leishmania donovani* | CP022617.1 | AACTAACGCT-**ATATAAGTA**TCAGTTTCTGTACTTTATTG |
| *Leishmania infantum* | AF097653.1 | AACTAACGCT-**ATATAAGTA**TCAGTTTCTGTACTTTATTG |
| *Blastocrithidia culicis* | DQ860204.1 | AACTAACGCT-**ATATTTGTT**ACAGTTTCTGTACTATATTG |
| *Blastocrithidia culicis* | DQ860203.1 | AACTAACGCT-**ATATTTGTT**ACAGTTTCTGTACTTTATTG |

configuration of *H. muscarum* was an intermediate between the tubulin array configurations of *T. brucei* and *L. major*.

## The predicted *Herpetomonas muscarum* proteome

Orthofinder [30] was used to identify orthologous proteins from other trypanosomatids in the predicted proteome of *H. muscarum*. For the analysis, protein coding genes from the following species were used: 9 *Trypanosoma* species/subspecies (*Trypanosoma brucei brucei*, *Trypanosoma brucei gambiense*, *Trypanosoma congolense*, *Trypanosoma cruzi*, *Trypanosoma evansi*, *Trypanosoma grayi*, *Trypanosoma rangeli*, *Trypanosoma theileri* and *Trypanosoma vivax*), 4 *Leishmania* species (*Leishmania braziliensis*, *Leishmania donovani*, *Leishmania infantum* and *Leishmania major*); 6 additional monoxenous trypanosomatids along with our *Herpetomonas muscarum* predictions (*Angomonas deanei*, *Leptomonas pyrrhocoris*, *Leptomonas seymori*, *Crithidia bombi*, *Crithidia expoeki*, *Crithidia fasciculata*). Finally, we included a free-living, non-trypanosomatid kinetoplastid, *Bodo saltans*, as an outgroup. From these 21 species 87.5% of genes were assigned to 12,701 orthogroups (for summary see Table 3, full orthogroups table S3 Table). We found 7,265 of these orthogroups contained *H. muscarum* genes. There were 45 orthogroups containing only *H. muscarum* genes, these groups contain 215 genes. Overall, 90.7% of *H. muscarum* predicted proteins were assigned to an orthogroup.

Orthofinder also produced a phylogenetic tree based on protein sequences from proteins in orthogroups which contained a single gene from every species used in the analysis (Fig 4A).

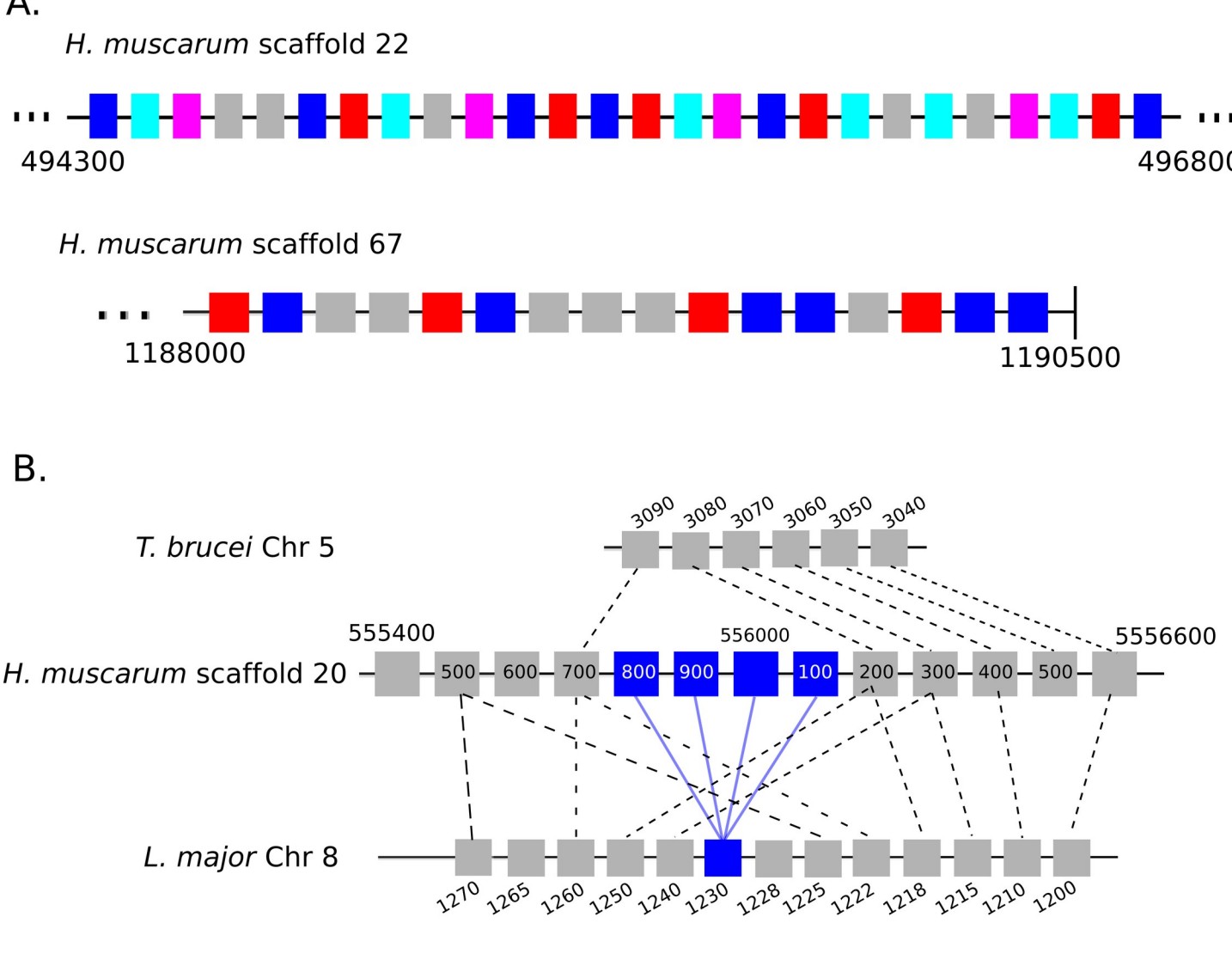

**Fig 3. A. Alternating tubulin arrays in *H. muscarum*.** Scaffolds 22 and 67 were found to have two loci containing alternating putative alpha (red) and beta (blue) tubulin genes. Several of these genes we predict to be tubulin pseudogenes (alpha—pink, beta—light blue) as they contain tubulin domains but also contain sequence consistent with non-LTR transposons. **B. A monotypic beta tubulin locus in *H. muscarum*.** Four copies of a putative beta tubulin (blue) were found in tandem on *H. muscarum* scaffold 20. This locus appears similar to the single copy beta tubulin locus on *L. major* chromosome 8 as the order of adjacent genes (grey) is conserved. We also see synteny with a locus in *T. brucei* on chromosome 5, however the beta tubulin gene is absent. Dotted lines indicate orthologous genes. Blue lines indicate orthologous beta tubulin genes.

This tree is consistent with others published for the trypanosomatids (Maslov *et al.*, 2013). Unsurprisingly *H. muscarum* shares more orthogroups with *L. major* (6,607) than *T. brucei*

**Table 3. Summary of Orthofinder analysis of 13 trypanosomatid genomes.** (Trypanosoma rangeli, Trypanosoma grayi, Trypanosoma brucei brucei, Trypanosoma brucei gambiense, Trypanosoma vivax, Trypanosoma congolense, Leishmania donovani, Leishmania major, Leishmania mexicana, Leptomonas pyrococcus, Leptomonas seymori, Crithidia fasciculata and Bodo saltans).

| | |
|---|---|
| Total number of genes | 212,664 |
| Number of genes in orthogroups | 186,070 |
| Number of unassigned genes | 26,594 |
| Percentage of genes in orthogroups | 87.50% |
| Number of unassigned genes | 12.50% |
| Number of orthogroups | 12,701 |
| Number of species-specific orthogroups | 313 |
| Number of genes in species-specific orthogroups | 4,212 |
| Percentage of genes in species-specific orthogroups | 2.0% |
| Mean orthogroup size | 14.7 |
| Median orthogroup size | 14 |
| Number of orthogroups with all species present | 9 |
| Number of single copy orthogroups | 0 |

(5,893)–which is more distantly related (Fig 4B). However, *H. muscarum* had slightly more orthogroups in common (6754) with the two *Leptomonas sp.* used in the analysis (Fig 4C). Finally, within the Leishmaniinae clade *H. muscarum* and two species of 'old world' *Leishmania*, *L. major* and *L. donovani*, shared 81.2% of their orthogroups (Fig 4D). A global examination of the patterns of gene family sharing between *H. muscarum*, and other trypanosomatid groups confirmed these patterns (Fig 5A). Most gene families, including most genes, are present in all of the groups, and another significant set of families is shared by all the trypanosomatid groups but missing from the outgroup, the free-living kinetoplastid *Bodo saltans*. These trypanosomatidae-specific gene families tend to be quite large, while many smaller gene families are specific to genera *Crithidia* and *Trypanosoma*, perhaps because of the more extensive taxon sampling of these lineages. There are exceptions, including some strikingly large gene families unique to trypanosomes, *Leishmania* and a number of other taxonomic groups (Fig 5B). Monoxenous trypanosomatids share many more genes families with *Leishmania* than *Trypanosoma*, and there are strikingly few families specific to the *Leishmania* lineage or any of the monoxenous parasites except *Crithidia*, explaining the strikingly similar predicted proteomes of *Leishmania* and *H. muscarum*.

We could not look in detail at all of the homology relationships between genes in this extensive comparison. We used a more focused OrthoFinder analysis to investigate specific groups of orthologues between *H. muscarum* and *T. brucei* genes of interest e.g. metabolic pathway genes, as *T. brucei* is the best-studied kinetoplastid at the molecular and cellular level. We summarise our findings in Table 4 (for full data see S4–S16 Tables) and discuss some of the orthologues of interest, including surprisingly 'missing' orthologues, below.

**Metabolism.** *H. muscarum* is missing sphingolipid (SL) biosynthesis genes SLS1-4, including the inositol phosphorylceramide synthase and two choline phosphorylceramide synthases. These genes are part of the same orthogroup from our analysis. Most of the *Trypanosoma* have 4 genes assigned to this orthogroup (with the exception of *T. cruzi* (2) and *T. vivax* (0)). However, other species used in this analysis had only 1 gene assigned to this orthogroup. Given that SLs are thought to be essential to eukaryotic membranes [31], this seemed surprising. However, *L. major* promastigotes do not require *de novo* SL synthesis and a mutant devoid of SLs was viable and replicated as log-phase promastigotes [32]. However, the SL-free mutant was unable to differentiate into a metacyclic stage *in vitro* and showed severe

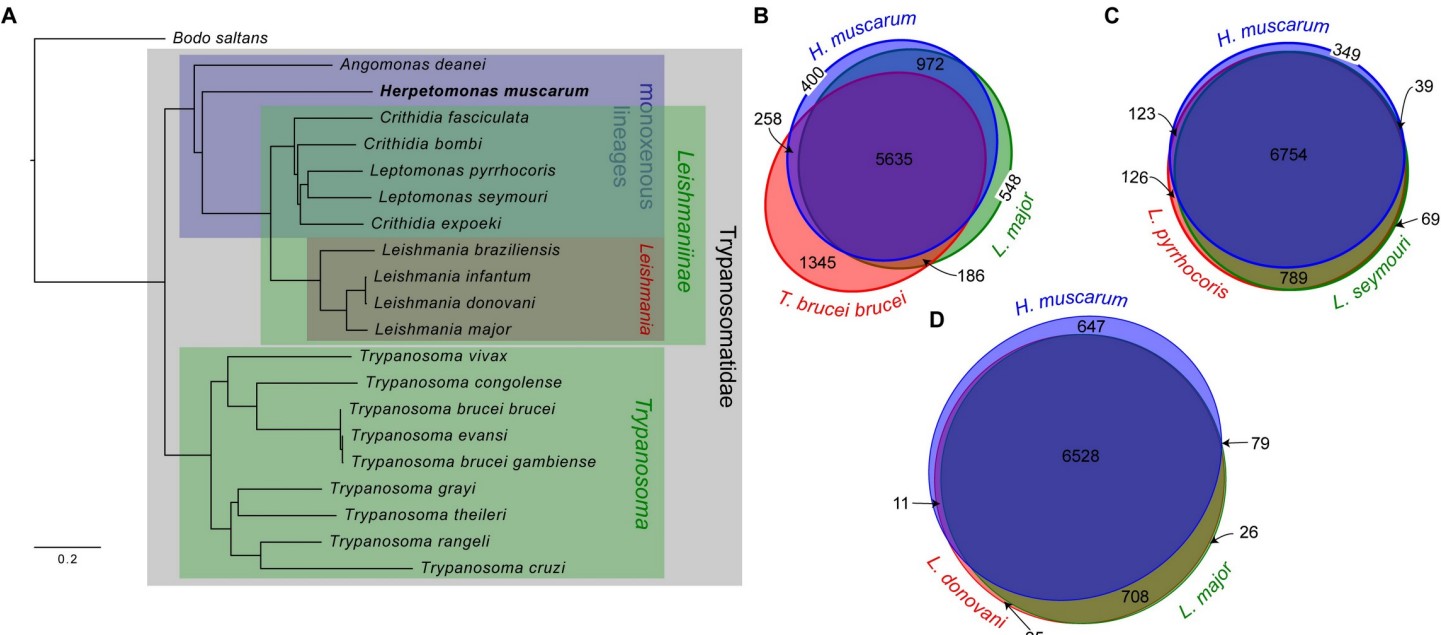

**Fig 4. Relationship between *H. muscarum* and other trypanosomatids. A.** Phylogeny based on all orthogroups containing a single gene from each species. Other panels show Venn-Euler diagrams in which the areas of each eliptical section are approximately proportional to the number of orthogroups shared by each of **(B)** *H. muscarum*, *L. major* and *T. brucei brucei*; **(C)** *H. muscarum*, *Leptomonas pyrrhocoris* and *L. seymouri* and **(D)** *H. muscarum*, *Leishmania donovani* and *L. major*. Diagram layouts were generated by EulerApe v2.0.3.

defects in vesicular trafficking. As such, like *L. major*, *H. muscarum* and the other species without a complete SLS pathway may rely on scavenging sphingolipids from the environment.

*H. muscarum* did not have orthologues for the carnitine O-acetyltransferase (CAT) (Tb927.11.2230) and L-threonine 3-dehydrogenase (Tb927.6.2790) genes of the acetate metabolism pathway. We were also unable to find an orthologue to these genes in other species from the Leishmaniinae clade used in the analysis. As such these genes may have been lost sometime after the group diverged from *Trypanosoma*.

Additionally, three *T. brucei* respiratory chain genes did not appear to have orthologues in *H. muscarum*, including mitochondrial NADH-ubiquinone oxidoreductase flavoprotein 2 (Tb927.7.6350), which had orthologues in all species used in the analysis apart from *H. muscarum*. Similarly, the only genomes in the analysis without an orthologue for the cytochrome c oxidase assembly protein (Tb927.10.3120) were *H. muscarum* and *Phytomonas EM1*. Given the importance of these genes, this likely indicates an important gap in the *H. muscarum* annotation. Finally, no orthologue was identified for the *T. brucei* alternative oxidase (AOX) (Tb927.10.7090) which is found in *Trypanosoma* and is upregulated in bloodstream forms. This oxidase is thought to enhance organisms ability to cope with stress associated with temperature change, infections and oxidative stress [33].

We also note that for several *T. brucei* genes there were multiple *H. muscarum* orthologues. Two of the most extreme examples of this being the high-affinity arginine transporter AAT13 [34, 35] and the endo-/lysosome-associated membrane-bound phosphatase 2 (MBAP2) which have 38 and 18 orthologous genes in *H. muscarum* respectively. The increased copy number of these genes, hints at their importance, though the reason for their high-copy number in *H. muscarum* is as yet unclear. AAT13 and MBAP2 have been shown to be highly upregulated in *Leishmania* after their ingestion by sand flies and in conditions of nutrient starvation [36, 37].

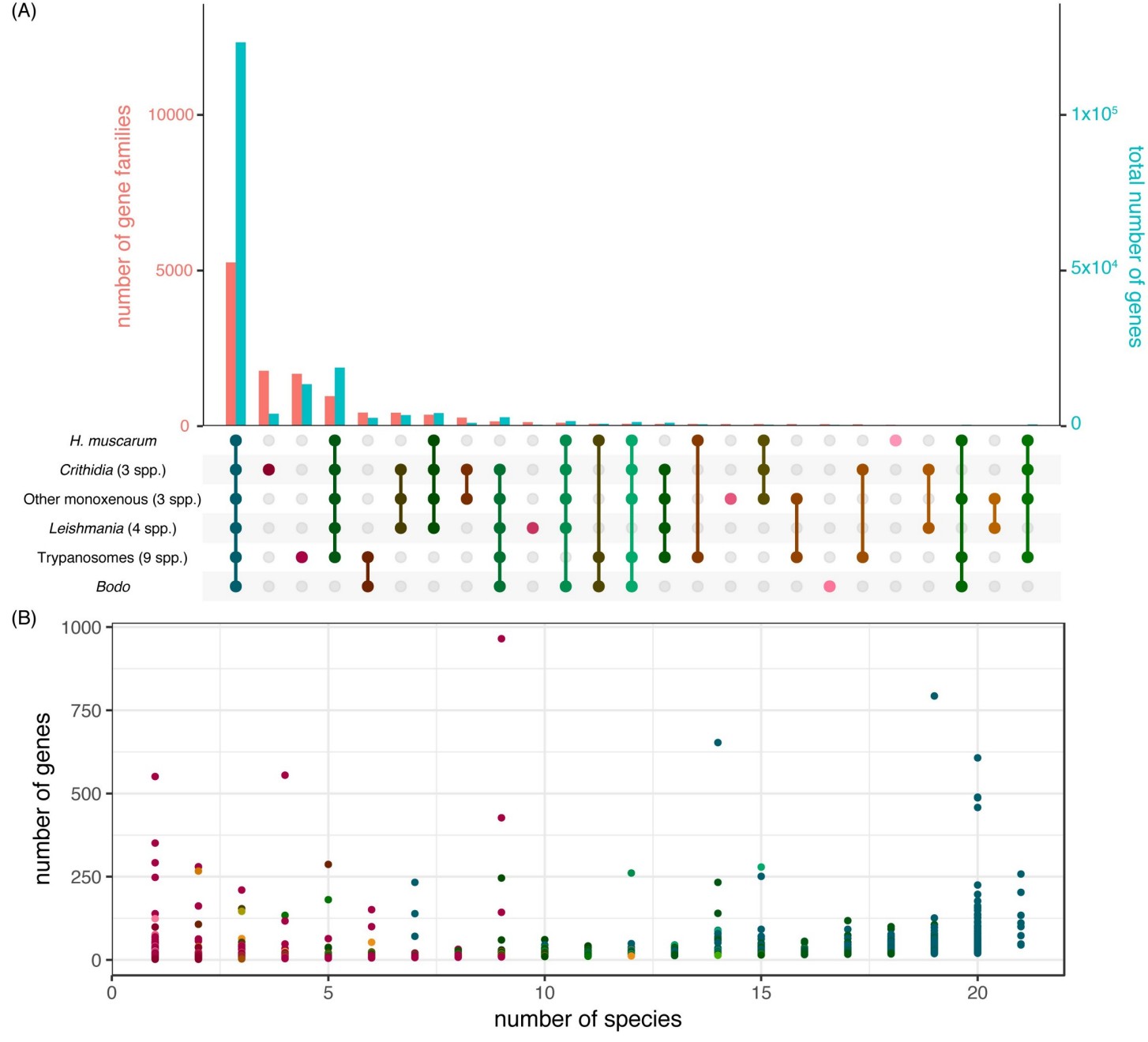

**Fig 5. A global view of gene family sharing between trypanosomatids. A.** The numbers of gene families (orthogroups; pink bars; values on left-hand y-axis) and the numbers of genes in those groups (blue bars; values on right-hand y-axis) with particular patterns of sharing between high-level groups in our Orthofinder data. Shading in the lower panel from pink to blue represents how widespread each set of families are, with pink representing families specific to one group and dark blue those families present in all groups. **B.** Scatterplot of gene family size against the number of species a family is present in, with each point representing a single gene family (families with less than 3 genes in total are excluded), and points coloured according to the number of higher-level taxonomic groups they are shared between, as in the lower part of panel **A.** [code to draw this diagram is a modified version of UpSetR].

Speculatively, the increased copy number of these genes may reflect the nutrient availability in *Herpetomonas*' environment/host(s).

**Differentiation.** RNA-binding proteins (RBPs) have emerged as key modulators of gene expression in trypanosomatids—particularly in the context of trypanosome development and

**Table 4. Summary of *H. musccarum* proteins orthologous to important *T. brucei* proteins.**

| | *H.muscarum* orthologues/*T. brucei* proteins | *T.brucei/L.major* without orthologues in *H. muscarum* |
|---|---|---|
| **METABOLISM** | | |
| Glycolysis | 44/45 | Tb927.10.4520 |
| Gluconeogenesis | 2/2 | n/a |
| Pentose phosphate pathways | 12/13 | Tb927.2.5800 |
| NADPH metabolism | 4/4 | n/a |
| Acetate metabolism | 14/17 | Tb927.11.2230, Tb927.8.2790, Tb927.6.2790 |
| TCA cycle | 17/17 | n/a |
| Mitochondrial carriers | 24/25 | Tb927.9.12140 |
| Respiratory chain | 79/82 | Tb927.7.6350, Tb927.10.7090, Tb927.10.3120 |
| Amino acid transporters | 31/31 | n/a |
| Lipid metabolism | 9/11 | Tb927.10.11930, Tb927.4.2700 |
| Leu-Isoleu-Val degradation | 22/23 | Tb927.4.2700 |
| Fatty Acid Biosynthesis | 14/14 | n/a |
| Sphingolipid biosynthesis | 7/11 | Tb927.9.9410, Tb927.9.9400, Tb927.9.9390, Tb927.9.9380 |
| Glycerophspholipid biosynthesis | 16/16 | n/a |
| GPI-N-glycosylation biosynthesis | 47/49 | Tb927.4.4200, Tb927.1.4830 |
| **DIFFERENTIATION AND DNA** | | |
| Quorum sensing | 32/35 | Tb927.4.3650, Tb927.11.2250, Tb927.11.11480 |
| Bloodstream to procyclic form differentiation | 10/12 | Tb927.10.10260, Tb927.10.11220 |
| Epimastigote meiosis | 4/5 | Tb927.9.15510 |
| RNA regulators of the life cycle | 18/18 | n/a |
| Proteins with RNA-binding annotation | 54/57 | Tb927.10.14950, Tb927.6.2550, Tb927.9.6870 |
| RNAi machinery | 5/5 | n/a |
| **PROTEIN KINASES** | 147/169 | Tb11.v5.0564, Tb11.v5.0644, Tb927.1.3130, Tb927.10.12480, Tb927.10.15880, Tb927.10.4940, Tb927.10.9980, Tb927.11.5150, Tb927.11.5860, Tb927.3.1850, Tb927.3.3920, Tb927.3.5650, Tb927.3.840, Tb927.4.4330, Tb927.5.4430, Tb927.7.4090, Tb927.9.12400, Tb927.9.12880, Tb927.9.1500, Tb927.9.1570, Tb927.9.16260, Tb927.9.2350 |
| **PHOSPHATASES** | 86/93 | Tb927.07.v5.1, Tb07.30D13.60, Tb927.10.4930, Tb927.11.11740, Tb927.11.4990, Tb927.11.5740, Tb927.8.8040 |
| **NUCLEAR PROTEOME** | | |
| Nuclear Pores | 27/27 | n/a |
| Exosome | 12/12 | n/a |
| Spliceosome | 56/59 | Tb927.10.7390, Tb927.9.6870, Tb927.3.1090 |
| Kinetochore | 30/34 | Tb927.10.6330, Tb927.11.1030, Tb927.5.4520, Tb927.9.13970 |
| **OTHER PROTEINS OF INTEREST** | | |
| GP63 | 14/15 | Tb927.11.7610 |
| Mucins | 8/11 | Tb927.8.7190, TcMUCII, Tb927.11.18610, Tb927.11.3400 |
| LPG biosynthesis | 20/29 | LmjF.14.1400, LmjF.02.0160, LmjF.02.0170, LmjF.02.0190, LmjF.02.0200, LmjF.02.0210, LmjF.02.0230, LmjF.35.0010, LmjF.25.2460, LmjF.31.3190, LmjF.36.0010, LmjF.02.0010, LmjF.21.0010, LmjF.07.1170, LmjF.34.0510, LmjF.02.0180, LmjF.02.0220, LmjF.05.1230, LmjF.19.650, LmjF.32.3900 |
| Trypanothione synthesis | 2/2 | LmjF.05.0350, LmjF.27.1870 |

differentiation [38]. Orthologues were found for 72/75 *T. brucei* RNA-binding proteins. RNA-binding proteins with no orthologues found in *H. muscarum* were: chromatin-remodelling-associated RRM2 (Tb927.6.2550) [39], the pre-RNA processing protein RBSR1 (Tb927.9.6870) [40] and a hypothetical RBP (Tb927.10.14950).

We have not observed differentiation in *H. muscarum* using 'classical' temperature/pH manipulations *in vitro* or during *D. melanogaster* infections. As such the 'completeness' of the *H. muscarum* RBP repertoire, relative to *T. brucei* which has multiple discrete forms, is of interest. Several of these proteins had multiple orthologues in *H. muscarum* including RBP10 (4 orthologues, Tb927.8.2780). RBP10 is known to be highly expressed in bloodstream forms of *T. brucei* and its overexpression in procyclics led to an increase of many bloodstream-form specific mRNAs, as well as transcripts associated with sugar transport, the flagellum and cyto-skeleton [41]. The role for this protein in *H. muscarum* is unclear, as it does not appear to have a *bona fide* vertebrate host, however given this proteins links to sugar transport, it may play a more general role in metabolism in *H. muscarum*. Comparisons of *H. muscarum* RBP expression levels/timings with other trypanosomatids may shed more light on their role in the cell and potentially why we do not observe differentiated forms for this species.

In addition to the RBPs, we were unable to find any orthologues for the hydrophilic acyl-ated surface proteins (HASPs) or small hydrophilic endoplasmic reticulum-associated proteins (SHERPs) which are associated with metacyclogenesis in *Leishmania*. We also note that the repressor of differentiation kinase 1 (RDK1, Tb927.11.14070) has 6 orthologues in *H. mus-carum*. In *T. brucei*, RDK1 acts with the PTP1/PIP39 phosphatase cascade to prevent uncon-trolled differentiation from bloodstream to procyclic form [42]. Given that *H. muscarum* is thought to be confined to insects, the presence of multiple copies of this gene which assists in maintaining a 'vertebrate' cell form in *T. brucei* is intriguing. It may be that this protein has an alternative role in *H. muscarum*.

**Surface proteins.**   No orthologues were found for the EP procyclins which are known to be expressed highly *T. brucei* procyclic whilst in the tsetse vectors and are thought to provide protection from the digestive enzymes in the insect midgut [43, 44]. As such *H. muscarum* likely relies on other surface proteins for protection in the insect midgut (see the transcrip-tomic data below).

The lipophosphoglycan (LPG) is an abundant component of the *Leishmania* cell surface and its importance during multiple stages of the *Leishmania* life cycle, including interactions with the insect gut epithelium, is well known [45, 46]. As such the prescence of LPG synthesis ezymes in *H. muscarum* is of great interest (see Table 4 and S9 Table). Single copy orthologues were found for the LPG biosynthesis-associated proteins GPI12/14 and LPG2-5. The β-galac-tofuranosyl transferases LPG-1, -1R and -1L were grouped together in a single orthogroup (orthogroup 32) which contained 12 *H. muscarum* orthologues. However, no orthologues could be found for the β-galactofuranosyl transferases LPG1G1-3 in *H. muscarum*, these genes were only found in *Leishmania* species and *L. pyrrocoris* in our analyses (orthogroup 7861). Orthogroup 32 contained genes from all species used in this analysis with the exception of the two *T. brucei* sub species. Speculatively, orthogroup 32 may represent a more ancient group of these enzymes, whilst orthogroup 7861 may be a more recent development within the *Leish-mania/Leptomonas* species.

The three *L. major* side chain arabinosyltransferases SCA1, 2 and L were grouped into a sin-gle orthogroup (orthogroup 886). This orthogroup consisted of only *Leishmania*, *Leptomonas* and *T. grayi* proteins. Similarly, the *L. major* side chain galactosyltransferases (SCG1-7) and related proteins (SCGR1-6) were grouped into a single orthogroup (orthogroup 60) which contained protein sequences from only *Leishmania* and *Leptomonas* suggesting these proteins may be Leishmaniiae specific.

Orthofinder was unable to find an orthologue to the major surface proteins of salivary gland forms of *T. brucei*—BARPs (bloodstream alanine-rich proteins). These GPI-anchored proteins required for tsetse salivary gland colonisation [47, 48]. Additionally, we do not find orthologues for the *T. brucei* metacyclic invariant surface proteins (MISPs) which are found

extending above the VSG coat in salivary gland metacyclic forms [49]. Given the proteins are crucial for salivary gland colonisation, the lack of copies in the *H. muscarum* genome may partially explain the inability of *H. muscarum* to colonise the salivary glands of *D. melanogaster*, instead infections are confined to the insect crop and gut [20].

Finally, the 13 *T. brucei* GP63 genes were grouped with 28 *H. muscarum* genes. GP63 is a major surface protease in *L. major* promastigotes. The comparatively high copy number of GP63 in *H. muscarum* may highlight its importance. Furthermore, GP63 has been implicated in *Leishmania* virulence [50], and as such these will be of interest in future studies.

**Nuclear proteome.** Kinetochore interacting protein 3 (KKIP3, Tb927.10.6700) and SR protein (Tb927.9.6870) had no orthologues in *H. muscarum* or other species from the Leishmaniiae clade used in the analysis and as such they appear to be *Trypanosoma* specific. RNAi of KKIP3 in *T. brucei* resulted in defects in DNA segregation and reduced population growth [51].

Additionally, *T. brucei's* kinetochore interacting protein 1 (KKIP1), PHF5-like protein (Tb927.10.7390) and U1 small nuclear ribonucleoprotein 24 kDa (Tb927.3.1090) had orthologues in all species used in the analysis apart from *H. muscarum*. Similar to KKIP3, RNAi knock down of KKIP1 caused defects in DNA replication, though in the case of KKIP these defects were more severe–resulting in the loss of entire chromosomes [51]. It is unclear if these genes have been lost in *H. muscarum* or this indicates a gap in the current annotation. Based on the importance of KKIP1 and the fact these genes have orthologues in all other species analysed, it is likely to be the latter.

Finally, *H. muscarum* appears to have a 'full set' of the *T. brucei* RNA interference pathway genes including an orthologue for TbARGO1 (Tb927.10.10850). Genes from this well-conserved (in metazoans) pathway have been lost in several trypanosomatids including: *L. major*, *L. donovani* and *T. cruzi* [52, 53, 54]. The loss of this pathway in these organisms has been linked to *Leishmania* RNA virus perturbation [54, 55]—though this has not been explicitly demonstrated. Further investigations to look for evidence of viruses akin to the LRVs in *H. muscarum* could test the link between RNAi and virus infection in trypanosomatids. The presence of a functional RNAi pathway has also been linked to transposon activity in *Leishmania*–with RNA-negative species lacking active transposable elements (TEs), and RNAi competent *L. braziliensis* harbouring several classes of active TEs [55, 56]. Given this, it is possible that the loss/lack of active TEs in *L. major* and *L. donovani* have lifted the requirement of the RNAi pathway to protect against TE-associated genomic perturbations. We did observe transcripts corresponding to the telomere associated transposable elements (TATEs) in all *H. muscarum* transcriptomes (see below). As such, there may also be an important link between RNAi and transposon activity in trypanosomatids.

### The *H. muscarum* transcriptome during *in vitro* culture

We first analysed the transcriptome of *H. muscarum* during *in vitro* axenic culture, specifically to compare log-phase and stationary phase cultures. Knowledge of the log-phase transcriptome was especially important as this was the 'pre-infection' transcriptome in our *Drosophila* infection model. By comparing the log-phase *H. muscarum* transcriptome with that of *H. muscarum* in flies we sought to identify genes important in the establishment of infection (see section below). The principal component analysis (PCA) plot (S2 Fig) shows that the first principal component is mostly capturing variation between distinct clusters of samples from log and stationary phase and explains 68% of the variance in these data. As expected, we found extensive differential expression between log-phase and stationary phase, with 4044 genes significantly differentially regulated (p-adjusted <0.05) (S16 Table). This is approximately a third of the genome but most changes in expression were modest, with only 264 genes

**Table 5. Significantly differentially regulated cyclins and cyclin-related kinases between stationary, and log phase *H. muscarum*.**

| Gene Name | *H. muscarum* orthologue ID | log2FoldChange | adjusted p-value |
|---|---|---|---|
| CRK4 | HMUS00195900.1 | 1.3 | 8.89E-10 |
| cyclin 11 | HMUS01322900.1 | 1.2 | 4.32E-05 |
| cyclin 2 | HMUS00751100.1 | 1.2 | 2.72E-19 |
| cyclin 4 | HMUS00787500.1 | 1.1 | 1.53E-17 |
| cyclin 7 | HMUS00475100.1 | 0.8 | 2.41E-14 |
| CRK10 | HMUS01143000.1 | 0.7 | 6.49E-09 |
| cyclin 5 | HMUS00580100.1 | 0.7 | 2.02E-12 |
| cyclin 10 | HMUS01323000.1 | 0.5 | 0.001 |
| CRK12 | HMUS00986000.1 | 0.3 | 0.015 |
| DNA-directed RNA polymerase III subunit, putative | HMUS00638800.1 | -0.3 | 0.032 |
| mitochondrial DNA polymerase I protein C | HMUS00828800.1 | -0.5 | 0.006 |
| mitochondrial DNA polymerase I protein D | HMUS00617400.1 | -0.5 | 0.018 |
| mitochondrial DNA polymerase I protein B, | HMUS01100200.1 | -0.6 | 0.007 |
| DNA polymerase alpha/epsilon subunit B | HMUS00740000.1 | -0.7 | 0.004 |
| DNA polymerase delta catalytic subunit | HMUS00566500.1 | -0.7 | 0.006 |
| CRK3 | HMUS00914500.1 | -1.0 | 1.06E-40 |
| cyclin 8 | HMUS00524500.1 | -1.0 | 1.40E-39 |

upregulated $\geq$ 2-fold in stationary phase cells and 811 downregulated $\geq$ 2-fold which we will discuss further below. GO enrichment analysis, using Ontologizer [57], did not identify any significantly enriched GO terms associated with differentially regulated genes. However only 62% of *H. muscarum* genes have associated GO terms. As such, we looked for enrichment in Pfam domains. There were 26 Pfam domains significantly enriched in the genes upregulated in stationary phase and 73 Pfam domains significantly enriched among downregulated transcripts (S17 Table), which we discuss further below.

**Cell cycle associated proteins.** The Pfam domain associated with cyclins was significantly enriched in genes upregulated in stationary phase cells. From this, we investigated the expression profiles of the cyclins, and their associated kinases. Eleven were found to be differentially regulated between the two cell populations (Table 5).

There was significant downregulation of the mitosis-associated cyclin 8, CRK3 and several mitochondrial DNA polymerase subunits in stationary phase cells. Knockdown of CRK3 in *T. brucei* is associated with a reduction in cell growth [58]. Furthermore, there was upregulation of the $G_1$-associated cyclins 7, 4 and 11. These observations reflect the observed reductions in cell replication at higher cell densities. Consistent with this, and with a reduction in cell growth, there were also significant reductions in transcripts for α- and β-tubulins, DNA polymerases and several protein synthesis-related genes including: 40S ribosomal subunits, 28S rRNAs and five putative elongation factor 2 genes. However, there was also upregulation of mitosis-associated cyclin 2 in the stationary phase cells. Cyclin 2 has two roles in *T. brucei* procyclics: cell cycle progression through G1 and the maintenance of correct cell morphology at the posterior end of the cell [59]. The CRKs 10 and 12, which were also upregulated in stationary phase cells, have been shown to interact with cyclin 2 and their knock-down results in growth defects [60]. CRK12 is also essential to survival of *T. brucei* in mice and its depletion by RNAi lead to defects in endocytosis, an enlarged flagellar pocket and abnormal kinetoplast localisation [61]. Given the relative abundance of many transcripts associated with reduced replication in stationary phase cells, the upregulation of cyclin 2 and its associated CRKs (10 and 12) may be more relevant to the maintenance of correct cell morphology than mitosis.

**Stress and metabolism.** Stationary phase (of growth) is associated with build-up of toxic waste products and fewer nutrients available per cell. It was therefore unsurprising that we observed transcriptional changes indicating metabolic change and nutrient starvation. Genes containing the Pfam domain associated with major autophagy marker ATG8 were significantly enriched in stationary phase transcripts (33 in total). Autophagy is a vital process for survival in nutrient poor environments and involves the segregation of the cell components to be recycled into double membrane-bound vesicles called autophagosomes. The requirement for increased amounts of membrane in autophagy, may partially explain the upregulation of fatty-acid synthesis related genes in stationary phase, as fatty acids are crucial components of cell membranes. Three lipases, two putative lipase precursor-like proteins, fatty-acyl-CoA Synthase 1 and putative fatty acid elongase (ELO) protein were upregulated upon entry into stationary phase. This is consistent with observations of *Trypanosoma cruzi* cultures [62].

Whilst the upregulation of autophagy-related genes is an indicator of cell stress, we also observed the downregulation of several genes with domains associated with responding to oxidative stress including: thioredoxin, glutathione S-transferase and alkyl hydroperoxide reductase (AhpC) and thiol specific antioxidant (TSA). As such, cells do not appear to be under significant oxidative stress. Other forms of stress, such as reduced nutrient availability or pH changes, may be driving the predicted increases in autophagy. Additionally, transcripts bearing the heat shock protein 60 HSP60 domain (PF00118) were also significantly enriched in the downregulated transcripts, which is another indicator of cell stress.

**Cell surface proteins.** Proteins sharing a domain (cl28643) with the variant surface protein (VSP) proteins of the *Giardia lamblia*, a flagellated intestinal pathogen, were highly represented among genes upregulated in stationary phase *H. muscarum*. In *G. lamblia*, these VSPs are integral membrane proteins rich in cysteine residues, often in CxxC repeats. They have a highly conserved C-terminal membrane spanning region which has a hydrophilic cytoplasmic tail with a conserved five amino acid CRGKA signature sequence, and an extended polyadenylation signal [63, 64]. One VSP, of hundreds in the *Giardia* genome, is expressed per *Giardia* cell and they are thought to protect the cells from proteolysis [65]. A similar strategy of surface protein expression is utilised by blood stage *T. brucei* cells [66]. This method of antigen switching plays a major role in immune system avoidance and survival in vertebrate hosts. In *H. muscarum* the VSP domain-containing genes are predicted, by Phobius [67], to encode proteins with 8–9% cysteine residues, and a single predicted transmembrane domain predicted at the C-terminus. Notably there were also ten VSP domain containing proteins downregulated upon entry into stationary phase.

In addition to the VSP domain containing genes, several other putative surface proteins were differentially regulated upon entry to stationary phase; two putative amastin genes were highly upregulated, and eight transcripts which encode for proteins with the cytomegalovirus UL20A protein domain (PF05984), were downregulated in stationary phase *H. muscarum* cells. The functions of proteins with UL20a domains, including the domains namesake, are largely unknown. Deletion of UL20a from the human cytomegalovirus genome resulted in reduced viral production in infected fibroblasts [68]. Further study will be required to elucidate the role of these proteins in trypanosomatids.

**Transcription.** The bias towards downregulated transcripts in the stationary phase cells as compared to log phase suggests a reduction of transcription and translation during stationary phase. Furthermore, five tRNA-synthase Pfam domains (PF00133.22, PF00749.21, PF00152.20, PF00587.25, PF01411.19) were significantly enriched in downregulated transcripts (chi-squared, $p < 0.05$) and RNA polymerase III subunits were also downregulated. Overall, transcriptomic changes associated with cell surface remodelling, autophagy and reductions in transcription were observed in cells entering stationary phase. Cyclin expression

patterns appear to suggest a bias in cells at G1 phase, as reported for *in vitro* culture of *T. brucei* procyclics [69].

## Transcriptome of *H. muscarum* inside *D. melanogaster* compared to *in vitro* culture

To identify potentially important *H. muscarum* genes during the infection of *D. melanogaster* we sought to analyse the transcriptome of the trypanosomatid over the course of infection by RNA-sequencing analysis. RNA was purified from infected flies at 6, 12, 18, and 54 hours post-ingestion of *H. muscarum*. The resulting RNAs were sequenced and mapped to the concatenated genomes of *D. melanogaster* and *H. muscarum*. Reads were later resolved to the corresponding species. Here we will discuss the resulting transcriptome of *H. muscarum*: the transcriptome of *D. melanogaster* after ingestion of *H. muscarum* in the same experiment was discussed elsewhere [20].

The number of reads which mapped to the *H. muscarum* genome ranged from 6949 to approx. 16.2 million reads per sample. At 6 hours post ingestion 40% of the total mapped reads were shown to map to *H. muscarum* (average of 3 biological replicates). This decreased to 20% in samples from 12 hours and 9% at 18 hours post ingestion. This correlates with the observed decrease in *H. muscarum* numbers as the parasite was cleared by *D. melanogaster* 18–54 hours post ingestion [20]. For differential expression analysis, only data up to 18 hours post infection was used as at 54 hours the number of sequencing reads mapping to the *H. muscarum* genome dropped below 1% of the total number of mapped reads (Fig 6).

Principal component analysis (PCA) shows that the first two principal components of variation in mRNAs between *H. muscarum* from in vitro culture and *H. muscarum* after ingestion by *D. melanogaster* explained 58% and 10% of the variance in these data (Fig 6B). The PCA plot shows a high degree of difference between the *in vitro* samples and samples isolated from infected flies. The level of change in expression was much higher than between the two *in vitro* conditions discussed above.

For the infections, log phase *H. muscarum* cultures were used to feed the flies. In order to identify transcriptomic changes in *H. muscarum* associated with being ingested by the fly, we compared the transcriptome of *H. muscarum* cells from log phase *in vitro* culture to the in-fly transcriptomes. Over a third of the genome, 4,633 genes, was significantly differentially regulated (Wald test, adjusted p-value < 0.05) between log phase axenic culture samples and samples from infected flies (S18 Table). Comparisons of gene expression between sequential time points over the course of infection revealed that there was a large initial transcriptomic change upon ingestion with 4662 genes differentially regulated between log phase culture and six hours post ingestion. This large initial transcriptomic shift was followed by more subtle transcriptomic changes between 6–12 (204 genes) and 12–18 hours (25 genes) (adjusted p-values < 0.05). Here we describe some of the changes in gene expression observed after ingestion and how these compare with other published transcriptome studies of trypanosomatids in their insect vectors including notable work by Inbar *et al.*, 2017 [37] on genes expression of four morphologically distinct *L. major* stages in a sand fly vector and Savage *et al.*, 2016 [70] on *T. brucei* in three tsetse fly tissues.

## *Herpetomonas muscarum* genes differentially regulated at six hours post-ingestion by *Drosophila melanogaster*

Approximately a third of the *H. muscarum* genome was found to be significantly differentially expressed between log phase axenic culture and six hours post ingestion by *D. melanogaster* (p < 0.05) (S19 Table). Of this subset, 640 genes had a fold change of ≥ 4 between the time

**A.**

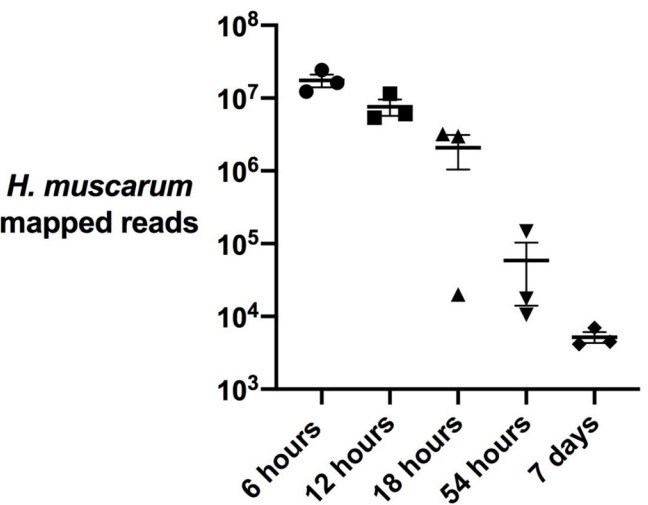

**B.**

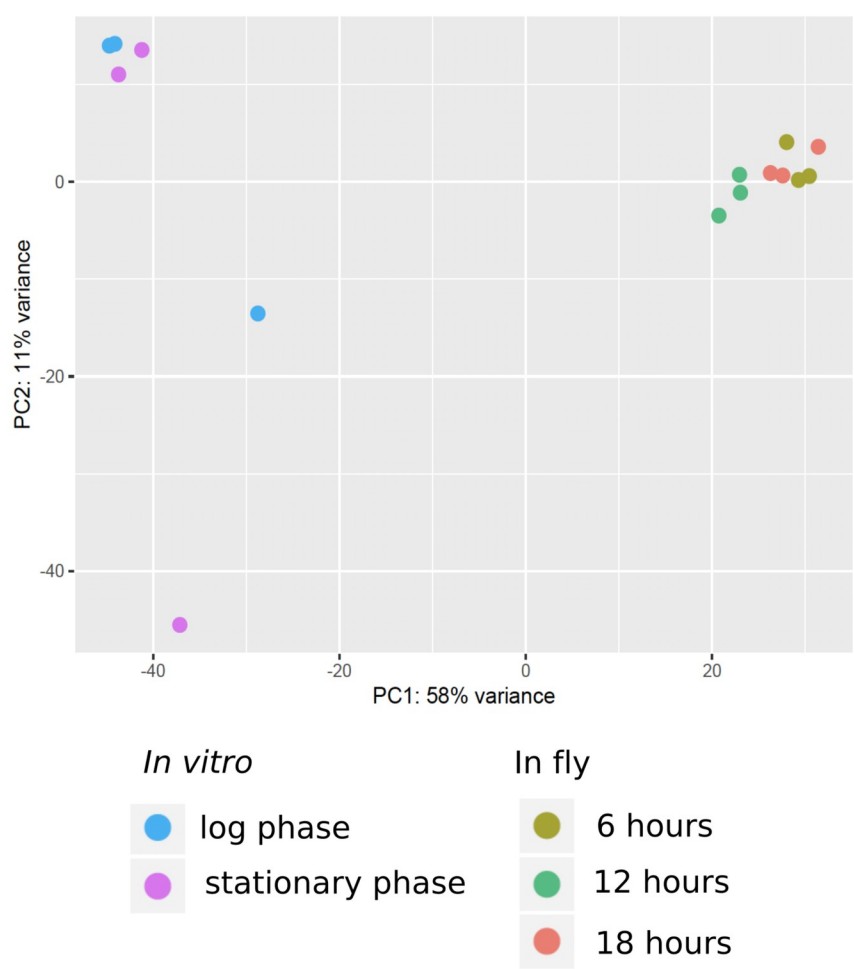

**Fig 6. A. RNA-seq reads extracted from infected flies (whole) which mapped to *H. muscarum* genome.** Error bars show the standard error of the mean. **B. Principal component analysis of differentially expressed *H. muscarum* genes in log phase culture vs. samples isolated from infection flies at 6, 12 and 18 hours post-ingestion.** There are two clear sample groupings (circled) which correspond to RNA from l in vitro culture log phase cells and RNA isolated from infected flies. Different shades of blue indicate the sample origin (n = 3 per condition).

points–highlighting the magnitude of the trypanosomatids response to ingestion. GO enrichment analysis, using Ontologizer [57], identified two significantly enriched GO terms in the 346 transcripts comparatively enriched at six hours post ingestion; OG0000045 (autophagosome assembly, p = 0.0014) and OG0003333 (amino acid transporters, p = 0.0002). Given the aforementioned lack of annotated GO terms in *H. muscarum*, we also looked at Pfam enrichment in the *H. muscarum* genes significantly upregulated upon ingestion by the fly. The top 15 represented Pfam domains in genes upregulated ≥ 4-fold at six hours post-ingestion are all significantly enriched compared to the full gene set (S20 Table). Additionally, there were several Pfam domains enriched in the downregulated transcripts, which we discuss further below.

**Leucine-rich repeat proteins.** The most represented Pfam domain in genes upregulated at 6 hours post ingestion were the leucine-rich repeat (LRR) domains. LRRs are primarily known to be involved in protein-protein and protein-glycolipid interactions and are the major domain of the *Leishmania* protein surface antigens (PSAs), which are known virulence factors. Ten of the upregulated LRR-containing genes encode orthologues of the *Leishmania* PSAs (Fig 7A). The predicted protein structures for 8/10 of these transcripts consists of a single transmembrane domain at the N-terminus, with the majority of the protein predicted to be on the external face of the cell (S21 Table). One transcript encodes a protein with no predicted transmembrane domains and could therefore be a secreted protein. The remaining transcript encodes a protein with two predicted transmembrane domains, with the region between these domains on the external face of the cell. Other upregulated LRR-containing transcripts are putative adenylate cyclases. These proteins also feature prominently in the *T. brucei* genes which are differentially regulated upon ingestion by tsetse [70]. These signalling proteins likely assist in the coordination of the trypanosomatids' responses to the environment with its vector.

**Cell surface genes.** Seven of the top fifteen genes, 21/346 overall, upregulated in *H. muscarum* at six hours post ingestion by *D. melanogaster* contained the *Giardia* variant-specific surface protein (VSP) domain (PF03302.13). These genes are members of three distinct orthogroups. A heatmap showing the normalised read counts for these genes across all samples is shown in Fig 7B. Transmembrane domain prediction tools [67, 71] predict a single transmembrane domain at the N-terminus in the majority of predicted protein sequence for these genes. However, there were also eight transcripts without predicted transmembrane domains, which are predicted to be secreted proteins. The majority of these putative surface antigens are 769–781 amino acids in length, have a single predicted transmembrane helix at residues 7–29 (S21 Table). As previously mentioned, many of these proteins are also upregulated by the cells upon entry into stationary phase, though not to the same levels. Additionally, several transcripts for VSP-containing proteins are downregulated in *H. muscarum* upon entry into the fly. These thirteen proteins are generally smaller than those upregulated at the same time point (95–501 amino acids) and tended to be part of orthogroup 11.

Thirty amastins, from 11 different orthogroups, were differentially regulated in *H. muscarum* at 6 hours post ingestion (Fig 7C). The majority (21) were upregulated upon entry into the fly, though 14 transcripts were also upregulated during stationary phase *in vitro* culture. Each orthogroup represented contained both up- and down-regulated genes. The function of this family of glycoproteins, are not well understood. In *Leishmania*, amastins are more

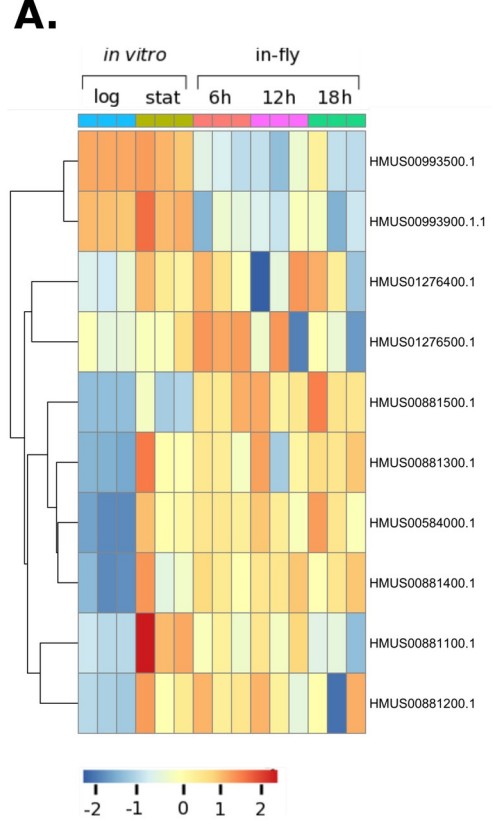

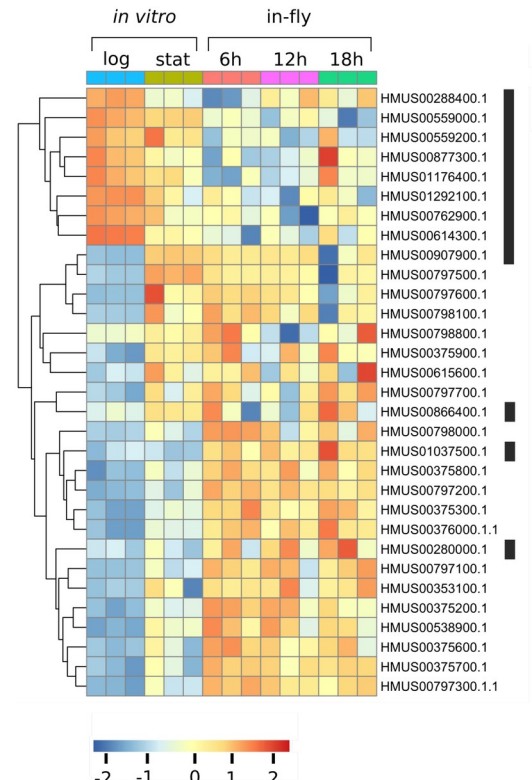

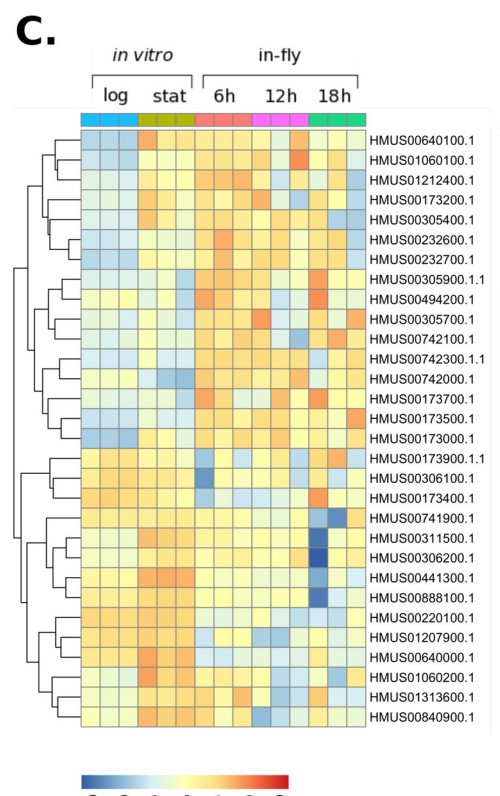

**Fig 7. Heat map of normalised, log transformed counts for differentially expressed *Herpetomonas muscarum* surface proteins. A.** *H. muscarum* orthologues to the *Leishmania* promastigote surface antigens. **B.** Transcripts encoding proteins with a *Giardia* variant surface protein (PF03302.13) domain. The black bar indicates the genes from orthogroup 11 which are mostly downregulated upon ingestion of *H. muscarum* by the fly. **C.** Differentially regulated *H. muscarum* amastin genes. Log = log phase axenic culture samples, Stat = stationary phase axenic culture samples. 6h = six hours post ingestion by *D. melanogaster*, 12h = twelve hours post ingestion by *D. melanogaster*, 18h = eighteen hours post ingestion by *D. melanogaster*.

commonly associated with macrophage-dwelling amastigote forms, where they are known to be important to both survival and virulence [72]. However, it has also been shown that β-amastins are upregulated during the insect stages of the life cycle in *T. cruzi* [73]. The *H. muscarum* amastins from orthogroup 18 share only 25–30% identity (across the whole sequence) to the two pairs of *T. cruzi* β-amastin alleles highlighted in this study. This may initially seem to be quite low, however the β-amastins have been shown to be highly divergent (18–25% identity) between *T. cruzi* strains [73]. Therefore, based on sequence alone, it is unclear which proteins may have parallel roles in the two trypanosomatid species.

Several other classes of surface protein genes were differentially expressed between log-phase axenic culture and six hours post-ingestion. Transcripts for proteins containing the Cytomegalovirus UL20A protein domain (PF05984) were significantly down regulated upon ingestion. Five of these genes were from orthogroup 11 –the same group as many of the down regulated VSP domain containing genes. Finally, sixteen (of the twenty-eight in the genome) *H. muscarum* orthologues to known *Leishmania* virulence factor, GP63, were significantly differentially regulated in the first six hours post ingestion by the fly. All but one of the differentially regulated GP63 orthologues were predicted to be GPI-anchored at the cell surface (GPI-SOM online tool) [74]. The exception, HMUS00892600.1, is predicted (THTMM v2.0) [71] to have a single transmembrane domain and for the majority of the protein to be cytosolic. Most GP63 transcripts were upregulated in *H. muscarum* after ingestion (log2 fold-changes 0.29–2.73), however two putative GP63 genes, HMUS01311000 and HMUS01311200, were downregulated with log2 foldchanges of -1.94 and -1.58 respectively.

**Stress-related genes.**   The insect gut is a hostile environment. The presence of digestive enzymes, changes in pH and the insect's gut microbiota make surviving a difficult challenge for any invading organisms. In correlation with this, a number of stress-associated genes and pathways are upregulated in *H. muscarum* upon entry into the fly. As previously mentioned, autophagy is an important process for survival in stressful conditions where fewer nutrients are available—such as in the midgut of an insect. Similar to observed in stationary phase axenic culture, twenty-six putative ATG8 genes were upregulated in *H. muscarum* at six hours post ingestion compared to log-phase axenic culture–suggesting extensive protein recycling is occurring in the cells. Additionally, 40 heat shock protein 83 genes were shown to be upregulated at six hours after ingestion. Heat shock proteins act as molecular chaperones which stabilise other proteins, help them to fold correctly and be regulated after damage in stressful conditions. The upregulation of these genes provides further evidence that these cells are in a stressed state.

**Metabolism.**   There was significant enrichment of putative amino acid, pteridine and sugar transporters in the upregulated transcripts. These included the amino acid transporters (AATs) orthologous to the *Leishmania* amino acid permease 3 (AAP3), AAT11, AAT12 and AAT20. AAP3 has been shown to be arginine specific and is linked to virulence in *L. donovani* infections in humans [75]. AAT11 is upregulated in during stress responses associated to purine starvation [76]. In *L. major*, AAP3 and AAT20 were strongly upregulated in the motile, gut-dwelling nectonomad forms [37]. These transporters have been shown to transport neutral amino acids across the cell membrane, notably proline and alanine, which can be used as alternative carbon sources by trypanosomatids and are abundant in insect vectors haemolymph.

Six putative pteridine transporters were also upregulated in *H. muscarum* at 6 hours post ingestion. Pteridines are needed by trypanosomatids to produce enzyme cofactors such as biopterin. *Leishmania* parasites are unable to synthesize their own pteridines [77] and as such must scavenge them from their environment. It is not currently known if *H. muscarum* is also a pteridine auxotroph, however like the *Leishmania* species, the cells appear to scavenge from the environment upon entry into the fly.

Several transcripts putatively involved in lipid metabolism were downregulated in *H. muscarum* following ingestion by *D. melanogaster*, including triglyceride lipases and members of the biotin/lipoate protein ligase (BLPL) family. This contrasts what has been observed in *L. major* in the midgut of sand flies where genes from these families were upregulated [37]. Therefore, whilst upregulation of pteridine and amino acid transporters appears to be a conserved trypanosomatid response to being ingested by insects, lipid metabolism during insect infection may differ between trypanosomatid genera.

**Gene expression-related transcripts.** Consistent with the differential expression of many genes upon entry into the fly, and therefore a predicted increase in chromatin remodelling and translation activity, there was upregulation of histones (2A, 3 and 4), RNA polymerase subunits 1 and 2, putative 40S/60S ribosomal proteins and putative 28S beta rRNAs in H. muscarum after ingestion by the fly. This result is consistent with what has been reported in *T. brucei* where the 40S and 60S ribosomal subunits were amongst the most highly upregulated genes in cells isolated from the midgut and proventriculus of *G. morsitans* [70].

**Cell cycle.** Upon ingestion by the fly there was strong upregulation of putative G1-associated cyclins 4, 7 and 11 as well as the G1 associated cyclin-related kinase 1 (CRK1) [58]. Cyclin 6, cyclin 8 and CRK9, which are associated with the G2/mitosis transition [59, 78], were slightly downregulated suggesting a reduction in cell replication at six hours post ingestion (Table 6). Consistent with this there was also downregulation of putative DNA polymerase kappa, the theta DNA polymerase subunit and mitochondrial DNA polymerase subunits. Furthermore meiosis-associated genes NBS1, Rad50 and SPO11 were also downregulated.

Given the apparent reduction replication rate in *H. muscarum* cells at six hours after ingestion, the upregulation of nine tubulin genes (3 alpha- and 6 beta-tubulins) is likely to accommodate the changes in cell morphology, rather than to produce new daughter cells. Tubulin upregulation is also observed in *T. brucei* isolated from the midgut and proventriculus of *Glossina morsitans* [70], though these cells are replicative–as such the 'motivation' for increased tubulin gene expression may be different.

**Differentiation and RNA-binding proteins.** It is well documented that (human) disease-causing trypanosomatids have several life-cycle stages within their respective vectors. Coordinated differentiation between these discrete stages requires a suite of RNA-binding proteins (RBPs) which regulate parasite gene expression [38]. Despite the lack of observed differentiated forms in infections of *D. melanogaster*, several differentiation associated-RBPs are differentially regulated in the trypanosomatid after infection including RBP10 and hnRNP F/H. These proteins have been shown to regulate gene expression in *T. brucei* blood-stream forms [41, 79]. RNAi knockdown of RBP10 in bloodstream trypanosomes resulted in the downregulation of a large number of bloodstream form mRNAs [41]. The same study showed that overexpression of the protein in procyclics led to an increase of many bloodstream-form specific mRNAs, including genes involved in sugar transport. This is likely owing to the fact blood is a glucose-rich environment and the cell will attempt to utilize this ready carbon source [80]. Three out of the four orthologues of *Tb*RBP10 were strongly (> 4-fold) upregulated in *H. muscarum* cells after ingestion by *D. melanogaster*. During feeding experiments sucrose is added to the *H. muscarum* culture media to encourage the flies to feed. As such these genes may be unregulated in response to increased sugars available in the environment.

**Table 6. Cell cycle-associated proteins differentially expressed in *H. muscarum* upon ingestion by *D. melanogaster*.** Fold changes shown are at 6 hours post ingestion compared to log phase axenic culture.

| Gene Name | *H. muscarum* orthologue ID | log2foldchange | adjusted p-value |
|---|---|---|---|
| cyclin 11 | HMUS01322900.1 | -3.31 | 6.54E-27 |
| cyclin 4 | HMUS00787500.1 | -1.15 | 3.62E-13 |
| CRK4 | HMUS00195900.1 | -0.95 | 3.46E-03 |
| CRK1 | HMUS01116400.1 | -0.84 | 9.95E-08 |
| CRK8 | HMUS00385600.1 | -0.49 | 2.32E-02 |
| cyclin 7 | HMUS00475100.1 | -0.44 | 2.21E-02 |
| cyclin 8 | HMUS00524500.1 | 0.36 | 1.94E-02 |
| mitochondrial DNA polymerase I protein D | HMUS00617400.1 | 0.57 | 9.16E-03 |
| cyclin 6 | HMUS00719100.1 | 0.74 | 2.14E-02 |
| cyclin 5 | HMUS00580100.1 | 0.85 | 9.34E-05 |
| CRK9 | HMUS01274200.1 | 0.87 | 1.45E-03 |
| DNA polymerase theta catalytic subunit | HMUS00097200.1 | 1.15 | 1.51E-07 |
| mitochondrial DNA polymerase I protein C | HMUS00828800.1 | 1.25 | 7.27E-09 |
| DNA polymerase kappa | HMUS01207400.1 | 1.36 | 4.93E-03 |
| CRK11 | HMUS00452900.1 | 1.46 | 8.20E-04 |
| CRK12 | HMUS00986000.1 | 2.07 | 6.68E-18 |

However, several other cell-cycle regulating RBPs associated with blood-stream form trypanosomes were also upregulated in *H. muscarum* after ingestion by the fly, including zinc-finger domain-containing RBPs ZC3H11 and ZC3H18. The former is essential in bloodstream-form trypanosomes and is involved in protection from heat shock, whilst depletion of ZC3H18 delayed blood stream form-to-procyclic differentiation in *T. brucei* [81, 82]. As such the situation may be more complex than solely metabolism-driven expression changes.

In addition to parallels with blood-stream form trypanosomes, transcripts for ALBA3/4 proteins (named for their 'acetylation lowers binding affinity' domain) were significantly downregulated in *H. muscarum* upon entry into the fly. In *T. brucei*, these proteins are expressed in all stages, except those found in the tsetse proventriculus. RNAi knockdown of these proteins in *T. brucei* axenic procyclics resulted in elongation of the cell body and repositioning of the nucleus and the kinetoplast to resemble the epimastigote cell-stage [83]. As such the reduction in ALBA3/4 transcripts suggests there may be parallels between trypanosomes during the latter stages of tsetse infection and *H. muscarum* during *D. melanogaster* infection.

Other differentially regulated RNA-binding proteins with as yet unclear roles in differentiation included: the essential gene expression regulation protein RBP42 and ZC3H12, a protein associated with differentiation [38].

### *Herpetomonas muscarum* genes differentially regulated between six- and twelve-hours post-ingestion by *Drosophila melanogaster*

There were 204 genes which were differentially regulated between six- and twelve-hours post ingestion (p-adjusted < 0.05), 161 of these had a fold change of ≥ 2 with just 31 genes upregulated at the latter timepoint (S22 Table). Hypothetical proteins lacking functional information dominated the highly upregulated genes. The most enriched transcript at 12 hours post ingestion encodes a putative surface protein, the top blastp hit for which was the *Giardia* variant-specific surface protein VSP136-4. This suggests VSP domain-containing proteins continue to be important throughout infection of the fly. Two DNA replication and repair associated transcripts were also upregulated at 12 hours post ingestion: an orthologue of *T. brucei* cell division

cycle protein 45 (CDC45), and tyrosyl-DNA phosphodiesterase-like protein. CDC45 is part of the CMG (Cdc45·Mcm2–7·GINS) complex which functions as a helicase during DNA replication [84] and may also play a role in DNA repair [85]. Furthermore, Tyrosyl-DNA phosphodiesterases are involved in the repair of topoisomerase-related DNA damage [86]. These observations indicate that *H. muscarum* cells are under genotoxic stress after ingestion by *D. melanogaster*.

### *Herpetomonas muscarum* genes differentially regulated between twelve- and eighteen-hours post-ingestion by *Drosophila melanogaster*

In the 23 genes found to be upregulated at 18 hours post ingestion (compared to at 12 hours, see S23 Table) genes involved in binding to damaged DNA (OG00033330) were significantly enriched. Only two of these transcripts were able to be assigned putative functions: eukaryotic replication factor A and a structure-specific endonuclease. This observation provides further evidence of genotoxic stress in *H. muscarum* after ingestion, as indicated by other upregulated DNA repair genes at 12 hours post-ingestion.

The most highly upregulated transcript at 18 hours post ingestion was an orthologue of the *L. major* UDP-galactose transporter LPG5B. This protein allows import of UDP-galactose into the golgi body where they are used to synthesize phosphoglycans. Capul *et al.,* (2007) showed that, in *L. major*, loss of LPG5B resulted in cells with defects in proteophosphoglycans (PPG) [87]. PPGs are known virulence factors and are found in membrane bound, filamentous and secreted forms. The viscous secreted PPG is thought to protect the *L. major* in the gut and may also force the fly to regurgitate the infective *Leishmania* cells into the bite wounds of vertebrates.

### *Herpetomonas muscarum* genes differentially regulated between stationary phase in vitro culture and in-fly samples

Comparisons between stationary phase in vitro culture and in-fly samples revealed 5102 differentially expressed genes (adjusted p-value < 0.05). Approximately 55% of the genes differentially regulated between *in vitro* and in-fly samples were the same for log phase vs in-fly and stationary phase vs. in-fly comparisons (Fig 8). However, 1639 genes were only significantly differentially regulated in stationary phase vs in-fly comparisons (S24 Table). Genes differentially regulated between log phase in vitro culture and in-fly samples have already been discussed, we will now outline the genes only differentially regulated when the transcriptomes of stationary phase in vitro samples of *H. muscarum* are compared with those after ingestion by *D. melanogaster*. Of the 1639 genes, 750 had a fold change of ≥ 2 –approximately a third of which were upregulated in *H. muscarum* after ingestion by *D. melanogaster*.

Half of the top ten in-fly enriched transcripts were TATE (telomere associated mobile elements) DNA transposons and among the most represented Pfam hits in the fly-enriched transcripts were reverse transcriptase (PF00078.27) and phage integrase (PF00589.22) domains (S25 Table). Though TATE DNA transposons comprise 1.32% of the *L. major* genome, very little is known about these transposable elements, other than that they contain a tyrosine recombinase [88]. It is possible that these transposable elements are more mobile in *H. muscarum* cells ingested by the fly. However, we predict that the overall level of transcription of cells in stationary phase cultures are reduced (vs. log phase, see above). As such, the comparative increase in TATE transposon transcription between stationary phase cells and *H. muscarum* from *Drosophila* may not be specifically a result of ingestion, but a reflection of general transcription levels in the two groups of cells.

As previously discussed, transcripts for several proteins containing a *Giardia* VSP domain are enriched in stationary phase compared to log phase in axenic culture. However, five were

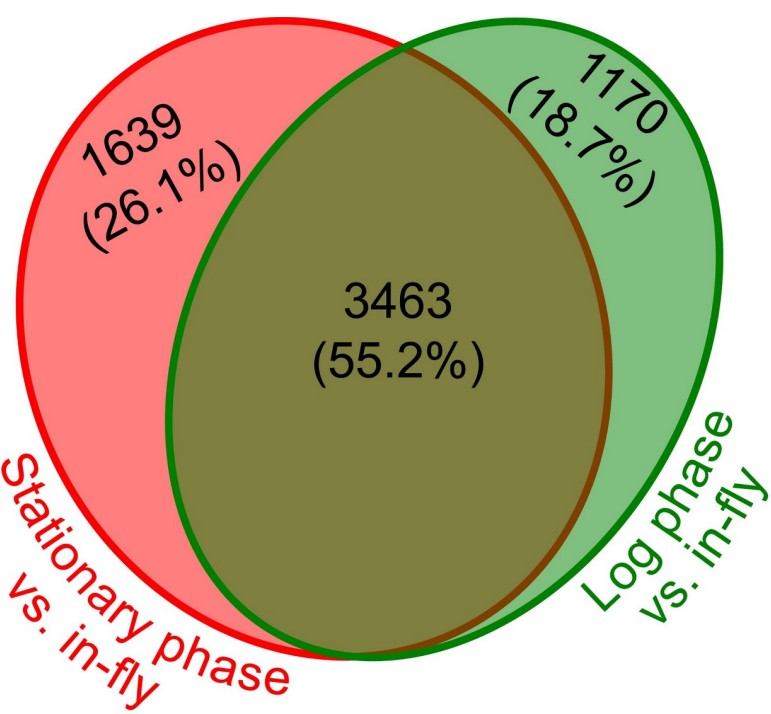

**Fig 8. Venn diagram showing the numbers of genes differentially expressed in *Herpetomonas muscarum* between two *in vitro* culture conditions and after ingestion by *Drosophila melanogaster*.**

shown to be even more abundant in the *H. muscarum* cells ingested by *D. melanogaster*. Two other putative surface antigens were also enriched in ingested *H. muscarum* which contained a domain similar to Cytomegalovirus UL20A glycoprotein and the domain of unknown function DUF4148.

Transcripts encoding for putative antioxidant proteins were significantly enriched in *H. muscarum* after ingestion by the fly. Enriched Pfam domains in the upregulated gene set included thioredoxin, glutathione S-transferase and alkyl hydroperoxide reductase (AhpC)/ thiol specific antioxidant (TSA) domains. Our previous work showed that the *D. melanogaster* response to *H. muscarum* ingestion included the production of reactive oxygen species [20], as such the upregulation of these antioxidant proteins is likely an attempt to cope with this insect immune response.

## Conclusion

Here we have described the genome and predicted proteome of the monoxenous trypanosomatid *H. muscarum* and characterised the transcriptome of the parasite both in culture and inside the gut of its natural host *D. melanogaster*. *H. muscarum* shows similarity in both genome structure and content to *Leishmania*, with significant synteny to *L. major* and sharing 80% of orthogroups with other members of the subfamily Leishmaniinae. While most *Herpetomonas* genes have orthologs in other trypanosomatids, a number of genes found elsewhere appear to have been lost in *Herpetomonas*, in particular genes associated with the specialised life stages of dixenous trypanosomatids. We might expect loss of some mammal-stage specific genes, such as HASPs, HERPs and sphingolipid synthesis genes important in metacyclic *Leishmania* cells, but more surprising might be the loss of genes expressed in insect stages such as BARPs and procyclins.

The transcriptome of *Herpetomonas* inside its insect host also showed strong parallels with the responses of *Leishmania* promastigotes inside the sand fly gut, in particular both parasites showing significant upregulation of PSAs and GP63 (this study; see ref. [37]). These proteins have been shown to be associated with virulence in *Leishmania* and are important for establishment of parasite infection in the midgut, and so for transmission. The extensive changes in transcript abundance of genes likely to be expressed on the cell surface during insect infections includes a number of gene families not known to be important in dixenous trypanosomatids (e.g. related to *Giardia* variant surface protein) implies that a dynamic cell surface may be a shared feature of trypanosomatid life cycles beyond dixenous groups [89], and that even more diversity of surface proteins may be present in the monoxenous trypanosomatids, supporting findings from free-living kinetoplastids. We also note that the majority of the genes showing changes in expression later in insect infections are hypothetical, including many hypothetical genes conserved with other trypanosomatids. This reflects similar findings in better-studied dixenous parasites [37, 70] and highlights how much we still have to learn about the interactions between trypanosomatids and their insect host.

In the wild, there is little data pertaining to the percentage of sand flies with established *Leishmania* infection in endemic regions. In this context, the parallel to the more accessible *Drosophila-Herpetomonas* system is important, as the genetic component of the parasite that influences midgut establishment is easier to determine. However, more work is needed to ascertain whether genes upregulated in *Leishmania* and those in *Herpetomonas* are truly functionally related. The limitation is the difference between the lifestyles of these insects. Most strikingly, female sand flies become infected with *Leishmania* during blood feeding, while *Drosophila* is never haematophagous. Nevertheless, sand flies are also plant feeders, so there is some overlap in the ecological niche as well as in their basic biology. The presence of trypanosomatids is another shared feature of the midgut landscape of these flies, and our data suggest that at least some aspects of the molecular interaction between flies and trypanosomatids may also be conserved.

## Materials and methods

### Herpetomonas muscarum culture

*H. muscarum* were cultured in supplemented BHI (3% brain heart infusion broth, 2.5mg/ml haemin, 1% FCS) and incubated at 28˚C. For most experiments, cells were maintained in a log phase of growth by splitting every 3 days.

### Infection of D. melanogaster (see reference 20)

For each independent infection of a group of 20–30 flies, $10^7$ *H. muscarum* cells were harvested from a 3 days-old culture (which showed the highest infectivity rate from our experience) and resuspended in 500ul 1% sucrose. The parasite solution was then transferred to a 21mm Whatman Grade GF/C glass microfibre filter circle (Fisher Scientific). Circles containing the parasite cells were placed into standard *Drosophila* small culture vial without any food. The flies used in the infections were 4–5 days old before they were starved overnight. After starvation, the flies were transferred to food vials that contained the Whatman circles with the parasite cells. After 6h of feeding, flies were moved and reared on standard yeast/molasses medium. At different time points post oral infection, infected flies were collected for downstream experiments and frozen at -80˚C for molecular analyses.

## DNA extraction for genome sequencing

Genomic DNA was extracted from 100 million *H. muscarum* cells from log phase cells from *in vitro* culture using the Norgen Biotek Genomic DNA extraction kit according to the manufacturer's instructions.

## RNA extraction for RNA-seq

8ml of *H. muscarum* promastigote culture at a density of 9.25 x$10^6$ cells per ml (measured by haemocytometer) was diluted 1:40 in supplemented BHI and divided between 4 tissue culture flasks. The immediate post-dilution density was 6.5 x$10^5$ cells per ml. The following day the cell density was measured to be 1.18 x $10^6$ cells per ml. 45ml was taken from each flask and the cells pelleted by centrifugation for 10 mins at 1000xg. The supernatant was discarded and the Norgen Biotek RNA Purification kit was (according to manufacturer's instructions) used to purify RNA from the cell pellet. This process was repeated for 5.3ml of the remaining culture three days later when the cell density was 1.21x$10^7$ cells per ml. The resulting RNA was eluted at concentrations 97–170 ng per μl with a 260/230 absorbance 1.86–2.19.

## Reference genome

To produce the reference genome Illumina and Pacific Biosciences sequencing platforms were used. For Illumina sequencing 1ug of genomic DNA was sheared into 300–500 base pair (bp) fragments by focused ultrasonication (Covaris Adaptive Focused Acoustics technology, AFA Inc., Woburn, USA). An amplification-free Illumina library was prepared [90] and 150 bp paired-end reads were generated on an Illumina MiSeq following the manufacturer's standard sequencing protocols [91]. For the Pacific Biosciences SMRT technology, 8 μg of genomic DNA was sheared to 20-25kb by passing through a 25mm blunt ended needle. A SMRT bell template library was generated using the Pacific Biosciences issued protocol (20 kb Template Preparation Using BluePippin(tm) Size-Selection System). After a greater than 7kb size-selection using the BluePippin(tm) Size-Selection System (Sage Science, Beverly, MA) the library was sequenced using P6 polymerase and chemistry version 4 (P6C4) on 6 single-molecule real-time (SMRT) cells [92].

The Pacific Bioscience reads were assembly with HGAP3 [93], with genome size parameter set to 25Mb, to produce 285 contigs. The obtained assembly was then corrected with ICORN2 [94], for five iterations. Using the Argus Optical Mapping System from OpGen, an optical map was generated from high molecular weight genomic DNA captured in agarose plugs and the restriction enzymes KpnI and BamHI. The data was analysed with associated MapManager and MapSolver software tools (http://www.opgen.com/products-services/argus-system). The optical map consisted of 37–38 chromosomes with approximately half being contiguous. With the information obtained from the optical map and REAPR [95], manual genome improvement was performed on the PacBio assembly to produce a final genome assembly of 181 contigs. Analysis of the frequency distribution of Kmers was performed using GenomeScope version 1.0 [22] with the kmer frequencies estimated using Jellyfish [96] using the default parameters suggested in the GenomeScope manual.

Transcriptomic libraries Poly-A mRNA was purified from total RNA using oligodT magnetic beads and strand-specific indexed libraries were prepared using the KAPA Stranded RNA-Seq kit followed by ten cycles of amplification using KAPA HiFi DNA polymerase (KAPA Biosystems). Libraries were quantified and pooled based on a post-PCR Agilent Bioanalyzer and 75 bp paired-end reads were generated on the Illumina HiSeq v4 following the manufacturer's standard sequencing protocols (as above).

### Data release

All sequencing data was submitted to the European Nucleotide Archive (ENA) under accession number ERP008869.

### Genome annotation

CRAM output files containing RNA sequencing reads from both *H. muscarum* in vitro culture and infected D. melanogaster were converted to fastq format and then mapped to the genome sequence using the next generation sequencing reads alignment package HISAT2 version 2.1.0 [97]. The mapped reads from each sample were assembled into transcripts with the Cufflinks package version 2.2.1[98] and merged to form a single transcript set for all reads. The Companion annotation tool [23] was then used to generate several genome annotation files based on the RNA sequencing transcriptomic evidence and pre-existing gene models from three other trypanosomatids–*L. braziliensis*, *L. major* and *T. brucei* (individual annotation statistics S26 Table).

### Orthofinder proteome analysis

The following proteomes were inputted into the Orthofinder script; *Trypanosoma brucei brucei* 927 v5.1 [24], *Trypanosoma brucei gambiense* DAL972 v3 [99], *Trypanosoma congolense* IL3000 [100], *Trypanosoma cruzi* (CL Brener) [27], *Trypanosoma evansi* STIB805 [101], *Trypanosoma grayi* ANR4 v1 [102], *Trypanosoma rangeli* SC_58 v1 [103], *Trypanosoma theileri* Edinburgh [104], *Trypanosoma vivax* Y486 [105], *Leishmania braziliensis* M2903 [56], *Leishmania donovani* BPK282 v1 [105], *Leishmania infantum* JPCM5 [56], *Leishmania major* Friedlin v6 [106], *Leptomonas pyrrhocoris* ASM129339v1 [11], *Leptomonas seymori* ASM129953v1 [107], *Crithidia bombi* [9], *Crithidia expoeki* [9], *Crithidia fasciculata* v14.0 [108], *Angomonas deanei* [8], *Phytomonas* EM1[109] and *Bodo saltans* v3 [110]. Where possible the above sequences were obtained from TriTrypDB v41 [111].

### RNAseq analysis in vitro culture

CRAM output files were converted to fastq format and then mapped to the concatenated *D. melanogaster* and *H. muscarum* genome sequences using the hisat2 [98] mapper. Mapped reads were then counted using HTseq-count (v. 0.10.0) [112] and differential expression analysed using the DESeq2 package in R [113].

### RNAseq analysis samples from whole flies

Total RNA of 8–10 flies at 6h, 12h, 18h post *H. muscarum* oral infection was extracted with total RNA purification kit from Norgen Biotek following the manufacturer's instruction. Each time point was repeated in three independent experiments. cDNA libraries were prepared with the Illumina TruSeq RNA Sample Prep Kit v2. All sequencing was performed on the Illumina HiSeq 2000 plaftform using TruSeq v3 chemistry (Oxford Gene Technology, OGT). All sequence was paired end and performed over 100 cycles. Read files (Fastq) were generated and then mapped to the concatenated *D. melanogaster* and *H. muscarum* genome sequences using the hisat2 mapper [98]. Mapped reads were then counted using HTseq-count (v. 0.10.0) [112] and differential expression analysed using the DESeq2 package in R [113].

## Supporting information

**S1 Fig. GenomeScope kmer profile and model for *H. muscarum* genome.**
(PDF)

**S2 Fig. Principal component analysis of differentially expressed *H. muscarum* genes in log phase culture vs. stationary phase culture. (A).** There are two clear sample groupings (circled) which correspond to RNA each condition (n = 3 per condition). Dark blue = log phase samples and light blue = stationary phase samples.
(PDF)

**S1 Table. Coordinates of putative strand switch regions in the *H. muscarum* genome.**
(XLSX)

**S2 Table. BLAST hits for the *Phytomonas serpens* spliced leader sequence in the *H. muscarum* genome.**
(XLSX)

**S3 Table. Alignment of intronic region of the mini-exon gene from several trypanosomatids of the Leishmaniiae clade and Protein orthogroups from Orthofinder analysis (in full).** The first 15bp of the intron sequences appear to be conserved across the clade with the sequence becoming more variable thereafter.
(XLSX)

**S4 Table. *H. musccarum* proteins orthologous to important *T. brucei* proteins: metabolism-associated proteins.**
(XLSX)

**S5 Table. *H. musccarum* proteins orthologous to important *T. brucei* proteins: Differentiation- and RNA-associated proteins.**
(XLSX)

**S6 Table. *H. musccarum* proteins orthologous to important *T. brucei* proteins: RNAi-associated proteins.**
(XLSX)

**S7 Table. *H. musccarum* proteins orthologous to important *T. brucei* proteins: Phosphatases.**
(XLSX)

**S8 Table. *H. musccarum* proteins orthologous to important *T. brucei* proteins: Protein kinases.**
(XLSX)

**S9 Table. *H. musccarum* proteins orthologous to important *T. brucei* proteins: GP63.**
(XLSX)

**S10 Table. *H. musccarum* proteins orthologous to important *T. brucei* proteins: Mucins.**
(XLSX)

**S11 Table. *H. musccarum* proteins without orthologues in other trypanosomatids.**
(XLSX)

**S12 Table. *H. musccarum* proteins orthologous to important *T. brucei* proteins: Kinetochore-associated proteins.**
(XLSX)

**S13 Table. *H. musccarum* proteins orthologous to important *T. brucei* proteins: Spliceosome-associated proteins.**
(XLSX)

**S14 Table.** *H. musccarum* **proteins orthologous to important** *T. brucei* **proteins: Exosome-associated proteins.**
(XLSX)

**S15 Table.** *H. muscarum* **proteins orthologous to important** *T. brucei* **proteins: Nuclear proteins.**
(XLSX)

**S16 Table.** *H. muscarum* **genes differentially expressed between log and stationary phase** *H. muscarum in vitro* **culture.**
(XLSX)

**S17 Table. Pfam domains significantly enriched in differentially regulated** *H. muscarum* **genes upon entry into stationary phase during axenic culture (vs. log phase).**
(XLSX)

**S18 Table.** *H. muscarum* **genes differentially expressed between log phase in vitro culture and after ingestion by** *D. melanogaster* **(all samples).**
(XLSX)

**S19 Table.** *H. muscarum* **genes differentially expressed between log phase in vitro culture and 6 hours after ingestion by** *D. melanogaster.*
(XLSX)

**S20 Table. Significantly enriched Pfam domains in differentially regulated** *Herpetomonas muscarum* **genes at six hours post ingestion by** *Drosophila melanogaster* **(vs log-phase axenic culture).** The table shows the top 10 represented Pfam domains in the significantly up- and downregulated genes. Chi-squared tests were performed to test for statistically significant enrichment of the Pfams frequency in upregulated genes vs. the Pfams in the whole genome.
(XLSX)

**S21 Table. Structural predictions for differentially expressed H. muscarum surface proteins.** Structural predictions were acquired using the TMHMM1.0 online tool (Krogh et al., 2001).
(XLSX)

**S22 Table.** *H. muscarum* **genes differentially expressed between 6 and 12 hours after ingestion by** *D. melanogaster.*
(XLSX)

**S23 Table.** *H. muscarum* **genes differentially expressed between 12 and 18 hours after ingestion by** *D. melanogaster.*
(XLSX)

**S24 Table. Table of genes differentially regulated between** *Herpetomonas muscarum* **after ingestion by** *Drosophila melanogaster* **vs. stationary phase axenic culture and not log phase axenic culture, p-adjusted < 0.05.**
(XLSX)

**S25 Table. The top 10 represented Pfam domains in** *Herpetomonas muscarum* **ingested by** *Drosophila melanogaster* **vs. stationary phase axenic culture.**
(XLSX)

**S26 Table. Summary statistics of three *H. muscarum* genome annotations using gene models from *L. major*, *L. braziliensis* and *T. brucei* and a maximal annotation which combines all three annotations.**
(XLSX)

# Acknowledgments

We thank the staff of the DNA pipelines at Wellcome Sanger Institute for sequencing and generating sequencing libraries.

# Author Contributions

**Conceptualization:** James A. Cotton, Petros Ligoxygakis.

**Data curation:** Megan A. Sloan, Karen Brooks, Thomas D. Otto, Mandy J. Sanders.

**Formal analysis:** Megan A. Sloan, Karen Brooks, James A. Cotton, Petros Ligoxygakis.

**Funding acquisition:** Petros Ligoxygakis.

**Investigation:** Megan A. Sloan, James A. Cotton.

**Methodology:** Thomas D. Otto, Mandy J. Sanders, James A. Cotton.

**Project administration:** Mandy J. Sanders.

**Resources:** James A. Cotton.

**Software:** Karen Brooks, Thomas D. Otto, James A. Cotton.

**Supervision:** James A. Cotton, Petros Ligoxygakis.

**Validation:** James A. Cotton.

**Visualization:** James A. Cotton.

**Writing – original draft:** Megan A. Sloan, Petros Ligoxygakis.

**Writing – review & editing:** James A. Cotton, Petros Ligoxygakis.

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
