## [Decision Letter · Decision Letter 0]

5 Sep 2019

Dear Dr Ligoxygakis,

Thank you very much for submitting your Research Article entitled 'Transcriptional and genomic parallels between the monoxenous parasite Herpetomonas muscarum and Leishmania' to PLOS Genetics.

The manuscript was fully evaluated at the editorial level and by two peer reviewers. We had hoped to secure a third reviewer but have now decided to proceed based on the reviews in hand. As you will see, both reviewers are enthusiastic about the work. There are some comments and suggestions that we ask you address in a minor revision. We therefore ask you to modify the manuscript according to the review recommendations before we can consider your manuscript for acceptance. Your revisions should address the specific points made by each reviewer.

[LINK]

Yours sincerely,

Won-Jae Lee

Guest Editor

PLOS Genetics

Gregory Barsh

Editor-in-Chief

PLOS Genetics

Reviewer's Responses to Questions

**Comments to the Authors:**

Reviewer #1: This manuscript entitled “Transcriptional and genomic parallels between the monoxenous parasite Herpetomonas muscarum and Leishmania” by Sloan et al describes the result of thoroughly conducted investigations on the genomic and transcriptomic features of Herpetomonas muscarum. The authors present the genome of H. muscarum, and further performed transcriptomics studies on axenic promastigotes in logarithmic and stationary phase as well as the parasites in the insect host of Drosophila melanogaster. Based on the sequencing result, an extensive analysis was carried out by comparing the features with dixenous trypanosomatids of Trypanosomes. Reported findings are not only of interest to the community of Herpetomonas and Drosophila, but may provide scientific insights to the researchers in the field of Trypanosomes.

There are few comments and suggestions.

1. In line 85, the authors stated that the parasites of interest “lack introns”. However, in the case of Trypanosoma brucei, Siegel et al (Nucleic Acids Res, 15:4946, 2010) reported an existence of two introns (Tb927.3.3160, a poly(A) polymerase and Tb927.8.1510, a DNA/RNA helicase). Thus, it is encouraged to state “virtually (or mostly) lack introns”. Plus, for H. muscarum, are those introns exists in the orthologs?

2. Trypanosomes are known to possess Base J (β-D-Glucopyranosyloxymethyluracil) acting as a RNA polymerase II transcription terminator. From this sequencing study, can the authors check the possible existence or signatures of Base J?

3. From line 281, the authors discuss some of the orthologues of interest based on OrthoFinder analysis (GP63, RNAi machinery, Leishmania RNA virus and etc were well discussed). On top of discussed orthologues, Leishmania and Trypanosoma parasites are known for other key features/virulence factors; trypanothione (with synthetase/reductase, Fairlamb et al, Science, 1985), lipophosphoglycan (Spath et al, Science, 2003), glycosome (Gualdrón-López et al, Int J Parasitol, 2012), acidocalcisome (Vercesi et al, Biochem J, 1994) and etc. The reviewer suggests the authors to discuss about these important features based on the genome sequencing data of H. muscarum.

4. Uniformity of Figure sub-numbering for corrections. (ex “A”, “A.”, “(A)” are all used in the figures).

5. For Table S26, two same files have been uploaded in Supplementary 15 and 16. There is no file uploaded for Table S25. Please check.

6. In the download system, Table S16 = supp7 and Table S17 = supp8. However, it is switched in the text. Please check.

7. As in Supplementary file 17, input of supporting Table # in the excel tab would help readers to easily follow the text and tables.

8. Abbreviations such as Leishmania major (L. major), Trypanosoma brucei (T. brucei) and etc should be maintained throughout the manuscript.

Reviewer #2: The present manuscript is descriptive investigation of the genomic and transcriptional parallels between the drosophila parasite Herpetomonas muscarum and Leishmania. At present, research on host- parasite interactions between Phlebotomine flies and Leishmania spp. remains sorely understudied due to a variety of constraints from routine colonisation of sand flies, adapted in vivo infection systems, but most importantly genetic tools to tease out the molecular mechanisms incurred during these unique host-parasite interactions in the midgut. Furthermore the notion of "midgut ecology" is now becoming more and more recognised and laboratories models for more in-depth studies will be necessary to better understand how microbes and parasites are interacting within this environment and with the mesenteron. The comparative results of your study are remarkable and convincingly support the promotion of the Drosophila-Hepetomonas model as a proxy for Phlebotomine flies and Leishmania spp. interactions and to better understand trypanosomid establishment in the dipteran insects midgut. I do appreciate the fact that you limit the focus of this comparison to this midgut compartment as it would be difficult to extrapolate further given that H. muscarum remains in the crop and midgut and has no interaction with the salivary glands as compared to dixenous trypanosomatid parasites.

Not only does the genome structure and organisation provide interesting parallels, but also the sometimes intriguing insights from the experimental transcriptional studies in vitro and in vivo. This however will require further investigation to understand the meaning and role of transcriptional flux and predicted proteome during infestation of the midgut (ex significance of tubulin array configurations or the significance of the surprising lack of the SLS1-4 gene for sphingolipid biosynthesis gene).

The paper is very well written and easy to follow. The data support the text of the manuscript. I did find one typographical error on line 313 ..."unregulated in Leishmania during sand flies and..." there is something missing here. Perhaps it should read, ""unregulated in Leishmania during sand fly infestation and..."

One aspect that you may want to consider in future studies is the idea of "pre-conditioning" of midgut ecology during larval midgut to adult midgut morphogenesis. It would be interesting to know if the larval microbiota could subsequently "facilitate" Heptomonas infections in the adult drosophila. Even though your studies were conducted under axenic conditions, it would be interesting to know how the carry-over of the larval micobiota or newly established adult microbiota are somehow affecting these early stages of parasite interaction with the mesenteron. Working on cave-dwelling sand flies we are intrigued by the presumed microbiota shift between the sand fly larval diet, which is essentially bat guano and organic materials from dead bats to the adult diet of blood and plant sap/nectar.

In conclusion this is a very important study to publish and I do hope it will encourage other drosophila researchers to embrace this model and shed more light on Phlebotomine flies and Leishmania interactions in the future.

**Have all data underlying the figures and results presented in the manuscript been provided?**

Reviewer #1: Yes

Reviewer #2: Yes

PLOS authors have the option to publish the peer review history of their article (what does this mean?). If published, this will include your full peer review and any attached files.

Reviewer #1: No

Reviewer #2: No

---

## [Editor Report · Decision Letter 1]

1 Oct 2019

Dear Dr Ligoxygakis,

We are pleased to inform you that your manuscript entitled "Transcriptional and genomic parallels between the monoxenous parasite Herpetomonas muscarum and Leishmania" has been editorially accepted for publication in PLOS Genetics. Congratulations!

Yours sincerely,

Won-Jae Lee

Guest Editor

PLOS Genetics

Gregory Barsh

Editor-in-Chief

PLOS Genetics

Comments from the reviewers (if applicable):

**Data Deposition**

http://datadryad.org/submit?journalID=pgenetics&manu=PGENETICS-D-19-01181R1

Press Queries

---

## [Editor Report · Acceptance letter]

1 Nov 2019

PGENETICS-D-19-01181R1 

Transcriptional and genomic parallels between the monoxenous parasite *Herpetomonas muscarum* and *Leishmania*

Dear Dr Ligoxygakis, 

We are pleased to inform you that your manuscript entitled "Transcriptional and genomic parallels between the monoxenous parasite *Herpetomonas muscarum* and *Leishmania*" has been formally accepted for publication in PLOS Genetics! Your manuscript is now with our production department and you will be notified of the publication date in due course.

With kind regards,

Nicholas White

PLOS Genetics

On behalf of:
